



**Biogenic volatile organic compound ambient mixing ratios and emission rates**
**in the Alaskan Arctic tundra**
**Hélène Angot[1], Katelyn McErlean[1], Lu Hu[2], Dylan B. Millet[3], Jacques Hueber[1], Kaixin Cui[1], Jacob Moss[1],**
**Catherine Wielgasz[2], Tyler Milligan[1], Damien Ketcherside[2], Marion Syndonia Bret-Harte[4], Detlev Helmig[1]**
[1]Institute of Arctic and Alpine Research, University of Colorado Boulder, Boulder, CO, USA.
[2]Department of Chemistry and Biochemistry, University of Montana, Missoula, MT, USA.
[3]Department of Soil, Water, and Climate, University of Minnesota, Minneapolis-Saint Paul, MN, USA.
[4]Institute of Arctic Biology, University of Alaska-Fairbanks, Fairbanks, Alaska, USA.
**Abstract**
Rapid Arctic warming, a lengthening growing season, and increasing abundance of biogenic volatile
organic compounds (BVOC)-emitting shrubs are all anticipated to increase atmospheric BVOCs in the
Arctic atmosphere, with implications for atmospheric oxidation processes and climate feedbacks.
Quantifying these changes requires an accurate understanding of the underlying processes driving BVOC
emissions in the Arctic. While boreal ecosystems have been widely studied, little attention has been paid to
Arctic tundra environments. Here, we report terpenoid (isoprene, monoterpenes, and sesquiterpenes)
ambient mixing ratios and emission rates from key dominant vegetation species at Toolik Field Station
(TFS; 68°38'N, 149°36'W) in northern Alaska during two back-to-back field campaigns (summers 2018
and 2019) covering the entire growing season. Isoprene ambient mixing ratios observed at TFS fell within
the range of values reported in the Eurasian taiga (0-500 pptv), while monoterpene and sesquiterpene
ambient mixing ratios were respectively close to and below the instrumental quantification limit (~2 pptv).
We further quantified the temperature dependence of isoprene emissions from local vegetation including
*Salix* spp. (a known isoprene emitter), and compared the results to predictions from the Model of Emissions
of Gases and Aerosols from Nature version 2.1 (MEGAN2.1). Our observations suggest a 180-215%
emission increase in response to a 3-4°C warming. The MEGAN2.1 temperature algorithm exhibits a close
fit with observations for enclosure temperatures below 30°C. Above 30°C, MEGAN2.1 predicts an isoprene
emission plateau that is not observed in the enclosure flux measurements at TFS. More studies are needed
to better constrain the warming response of isoprene and other BVOCs for a wide range of Arctic species.





## 1. Introduction

As a major source of reactive carbon to the atmosphere, biogenic volatile organic compounds
(BVOCs) emitted from vegetation play a significant role in global carbon and oxidation cycles
(Fehsenfeld et al., 1992). Global emission estimates of BVOCs are in the range of 700-1100 TgC
per year, ~70-80% of which corresponds to terpenoid species: isoprene, monoterpenes (MT), and
sesquiterpenes (SQT) (Guenther et al., 1995, 2006; Sindelarova et al., 2014). Despite their
relatively short atmospheric lifetimes (1 hour to 1 day for terpenoids), BVOCs affect climate
through their effects on the hydroxyl radical (OH, which dictates the lifetime of atmospheric
methane), tropospheric ozone ($O_3$, a key greenhouse gas), and aerosols (which influence radiative
scattering) (Arneth et al., 2010; Fuentes et al., 2000; Peñuelas and Staudt, 2010). The oxidation of
those BVOCs also drives the formation of secondary organic aerosols (SOA) through both gas-
and aqueous-phase mechanisms (Carlton et al., 2009; Lim et al., 2005). The potential for increased
SOA formation, expected to result in climate cooling (Kulmala et al., 2004), complicates the
climate feedbacks of BVOC emissions (Tsigaridis and Kanakidou, 2007; Unger, 2014).
Global models of BVOC emissions assume minimal emissions from the Arctic due to low leaf
area index and relatively cold temperatures (Guenther et al., 2006; Sindelarova et al., 2014).
However, this assumption relies on few observations and has been increasingly challenged by field
data (Tang et al., 2016). Recent measurements have revealed significant BVOC emissions from
Arctic tundra and vegetation, including *Sphagnum* mosses, wetland sedges, and dwarf shrubs
(Ekberg et al., 2009, 2011; Faubert et al., 2010; Holst et al., 2010; Lindfors et al., 2000; Potosnak
et al., 2013; Rinnan et al., 2011; Schollert et al., 2014; Tiiva et al., 2008). These results are of
importance because BVOC emissions are expected to increase in the Arctic due to climate
warming and associated vegetation and land cover change (Faubert et al., 2010; Potosnak et al.,
2013; Rinnan et al., 2011; Tiiva et al., 2008). Long-term field warming studies have shown strong
increases in BVOC emissions from shrub heath (Michelsen et al., 2012; Tiiva et al., 2008).
Furthermore, the temperature dependence of Arctic BVOC fluxes appears to be significantly
greater than for tropical and subtropical ecosystems (Holst et al., 2010; Rinnan et al., 2014), with
up to 2-fold increases in MT emissions and 5-fold increases in SQT emissions by subarctic heath
for a 2°C warming (Valolahti et al., 2015). Similarly, Kramshøj et al. (2016) and Lindwall et al.
(2016) examined the response of BVOC emissions to an experimental 3-4°C warming and reported





a 260-280% increase in total emissions. Together, the above results emphasize the strong
temperature sensitivity of BVOC emissions from Arctic ecosystems.
Changing BVOC emissions in the Arctic due to climate and land cover shifts can thus be expected
to perturb to the overall oxidative chemistry of the region, and to further affect climate through
various feedback mechanisms. Quantifying these changes requires an accurate understanding of
the underlying processes driving BVOC emissions in the Arctic. While BVOC ambient mixing
ratios and emission rates have been studied in boreal ecosystems, less attention has been paid to
Arctic tundra environments (Lindwall et al., 2015). Here, we report BVOC ambient mixing ratios
and emission rates at Toolik Field Station (TFS) in the Alaskan Arctic. This study builds on the
previous isoprene study at TFS by Potosnak et al. (2013), while also providing a major step forward
from that work. In particular, we present the first continuous summertime record of ambient
BVOCs (including isoprene and MT) and their first-generation oxidation products in the Arctic
tundra environment. We further compare the observed temperature dependence of isoprene
emissions with predictions from the Model of Emissions of Gases and Aerosols from Nature
version 2.1 (MEGAN2.1), a widely used modeling framework for estimating ecosystem-
atmosphere BVOC fluxes (Guenther et al., 2012). Due to increasing shrub prevalence across
northern Alaska (Berner et al., 2018; Tape et al., 2006), as well as the Eurasian (Macias-Fauria et
al., 2012) and Russian Arctic (Forbes et al., 2010), the results of this study have significance to
tundra ecosystems across a vast region of the Arctic.
**2.  Material and Methods**
**2.1 Study site**
This study was carried out at TFS, a Long-Term Ecological Research (LTER) site located in the
tundra on the north flanks of the Brooks Range in northern Alaska (68°38'N, 149°36'W; see
Fig.1). Vegetation speciation and dynamics, and their changes over time, have been well
documented at the site. *Betula* (birch) and *Salix* (willow) are the most common deciduous shrubs
(Kade et al., 2012). Common plant species include *Betula nana* (dwarf birch), a major player in
ongoing Arctic greening (Hollesen et al., 2015; Sistla et al., 2013), *Rhododendron tomentosum*
(formerly *Ledum palustre*; Labrador tea); Vaccinium vitis-idaea (lowbush cranberry), *Eriophorum*
*vaginatum* (cotton grass), *Sphagnum angustifolium* (peat moss), *Alectoria ochroleuca* (witches
hair lichen), and many other perennial species of Carex, mosses, and lichens. Vegetation cover at





this site is classified as tussock tundra (see Fig.1), which is the most common vegetation type in
the northern foothills of the Brooks Range (Elmendorf et al., 2012; Kade et al., 2012; Shaver and
Chapin, 1991; Survey, 2012; Walker et al., 1994).
Emission measurements and atmospheric sampling were conducted from a weatherproof
instrument shelter located ~350 m to the west of TFS (see Fig.S.I.1). Winds at TFS are
predominantly from the southerly and northerly sectors (Toolik Field Station Environmental Data
Center, 2019), minimizing any influence from camp emissions at the site. Two field campaigns
were carried out:  the first from mid-July to mid-August 2018, and the second from mid-May to
the end of June 2019. These two back-to-back campaigns cover the entire growing season (Sullivan
et al., 2007).
**2.2 Ambient online measurements of BVOCs and their oxidation products**
2.2.1   Gas chromatography and mass spectrometry with flame ionization detector

(GC-MS/FID)

An automated GC-MS/FID system was deployed for continuous measurements of atmospheric
BVOCs at ~2-hour time resolution during the 2018 and 2019 field campaigns. In addition, the
system was operated remotely following the 2018 campaign (through September 15[th]) to collect
background values at the beginning of autumn. Air was pulled continuously from an inlet on a 4
m meteorological tower located approximately 30 m from the instrument shelter (Van Dam et al.,
2013). Air passed through a sodium thiosulfate-coated $O_3$ scrubber for selective $O_3$ removal – to
prevent sampling losses and artifacts for reactive BVOCs (Helmig, 1997; Pollmann et al., 2005) –
and through a moisture trap to dry the air to a dew point of -30°C. Analytes were concentrated on
a Peltier-cooled (-40°C) multistage adsorbent trap. Analysis was accomplished by thermal
desorption and injection for cryogen free GC using a DB-1 column (60 m × 320 µm × 5 µm) and
helium as carrier gas. The oven temperature was set to 40°C for 6 minutes, then increased to 260°C
at 20°C/min, and held isothermally at 260°C for 13 minutes. The column flow was split between
an FID and a MS for simultaneous quantification and identification. Blanks and calibration
standards were regularly injected from a manifold. Isoprene (*m/z* 67 and 68), methacrolein
(MACR) and methylvinylketone (MVK) (*m/z* 41, 55, and 70), MT (*m/z* 68, 93, 121, and 136), and
SQT (*m/z* 204, 91, 93, 119, and 69) were identified and quantified using the MS in selected ion-
monitoring mode (SIM). The response to isoprene was calibrated using a primary gas standard



supplied by the National Physical Laboratory (NPL), certified as containing 4.01±0.09 ppb of
isoprene in a nitrogen matrix. The analytical uncertainty for isoprene was estimated at 16 % based
on the certified uncertainty of the standard and on the repeatability of standard analysis throughout
the campaigns. Instrument responses for MACR, MVK, α-pinene, and acetonitrile were calibrated
with multi-component standards containing 1007 ppb MACR, 971 ppb MVK, 967 ppb α-pinene,
and 1016 ppb acetonitrile (Apel-Riemer Environmental Inc., Miami, FL, USA) dynamically
diluted into a stream of ultra-zero grade air to ~3 ppb. Quantification of other terpenoid compounds
was based on GC peak area (FID response) plus relative response factors using the effective carbon
number concept (Faiola et al., 2012; Scanlon and Willis, 1985). The limit of quantification (LOQ)
was ~2 pptv (pmol/mol by volume). In order to monitor and correct for long-term trends in the
detection system, including detector drift and decreasing performance of the adsorbent trap, we
used peak areas for long-lived chlorofluorocarbons (CFCs) that were monitored in the air samples
together with the BVOCs as an internal reference standard. The atmospheric trace gases $CCl_3F$
(CFC-11) and $CCl_2FCCl_2F_2$ (CFC-113) are ideal in this regard because they are ubiquitous in the
atmosphere and exhibit little spatial and temporal variability (Karbiwnyk et al., 2003; Wang et al.,

2000).

2.2.2    Proton-Transfer-Reaction Time-of-Flight Mass-Spectrometry (PTR-ToF-MS)

During the summer 2019 campaign, isoprene mixing ratios in ambient air were also measured by
PTR-ToF-MS (model 4000, Ionicon Analytik GmbH, Innsbruck, Austria). The sample inlet was
located on the 4 m meteorological tower, right next to the GC-MS/FID inlet. In brief, ambient air
was continuously pulled through the PTR-ToF-MS drift-tube, where VOCs with proton affinities
higher than that of water (>165.2 kcal/mol) were ionized via proton-transfer reaction with primary
$H_3O^+$ ions, then subsequently separated and detected by a time-of-flight mass spectrometer (with
a mass resolving power up to 4000). At TFS, the PTR-ToF-MS measured ions from 17–400 $m/z$
every 2 minutes. Ambient air was drawn to the instrument at 10–15 L/min via ~30 m of 1/4" O.D.
PFA tubing maintained at ~55°C, and then subsampled by the instrument through ~100 cm of
1/16" O.D. PEEK tubing maintained at 60°C. The residence time from the inlet on the 4 m
meteorological tower to the drift-tube was less than 5 seconds. Instrument backgrounds were
quantified approximately every 5 hours for 20 minutes during the campaign by measuring VOC-
free air generated by passing ambient air through a heated catalytic converter (375 °C, platinum





bead, 1 % wt. Pt, Sigma Aldrich). Calibrations were typically performed every 4 days via dynamic
dilution of certified gas standard mixtures containing 25 distinct VOCs including isoprene (Apel-
Riemer Environmental Inc., Miami, FL, USA). Here, we report isoprene mixing ratios to inter-
compare with GC-MS measurements; other species will be reported in future work. The
measurement uncertainty for isoprene is ~25%, which includes uncertainties in the gas standards,
calibration method, and data processing.

2.2.3    Instrument inter-comparison

Figure S.I.2 shows a comparison of the GC-MS and PTR-ToF-MS isoprene mixing ratios in
ambient air. With a correlation coefficient of 0.93 and a linear regression slope of 0.7-1.0, the two
measurements agreed within their combined measurement uncertainties, in line with earlier inter-
comparison studies (e.g., Dunne et al., 2018; de Gouw et al., 2003). Similarly, we found a
correlation coefficient of 0.96 between GC-MS and PTR-ToF-MS MVK+MACR mixing ratios
(not shown). The good agreement between these two independent techniques gives us confidence
that the ambient air results presented here are robust.

**2.3 Ambient air vertical profiles**

Vertical isoprene mixing ratio profiles were obtained using a 12-foot diameter SkyDoc tethered
balloon. A total of eight vertical profiles were performed at ~3-hour intervals between 12:30 pm
Alaska Standard Time (AST) on June 15, 2019 and 11:00 am AST on June 16, 2019. Sampling
packages were connected to the tether line such that resulting sampling heights were ~30, ~100,
~170, and ~240 m above ground level. One identical sampling package was deployed at the
surface. Each sampling package contained an adsorbent cartridge for sample collection (see below)
connected to a downstream battery-powered SKC pocket pump controlled using a mechanical
relay, a programmable Arduino, and a real-time Clock. Once the balloon reached its apex (~ 250-
300 m a.g.l.), the five pumps were activated simultaneously and samples collected for 30 minutes.
At the end of the 30-min sampling period, the balloon was brought back down. The adsorbent
cartridges were prepared in house using glass tubing (89 mm long × 6.4 mm outer diameter, 4.8
mm inner diameter), and loaded with Tenax-GR and Carboxen 1016 adsorbents (270 mg of each),
following established practice (Ortega and Helmig, 2008 and references therein). An inlet ozone
scrubber was installed on each cartridge to prevent BVOC sampling losses. Field blanks were
collected by opening a cartridge (with no pumped airflow) during each balloon flight. Following





collection, samples were stored in the dark at ~4°C until chemical analysis. Samples were analyzed
at the University of Colorado Boulder following the method described in S.I. Section 1. Our
previous inter-comparison of this cartridge-GC-MS/FID method with independent and concurrent
PTR-MS observations showed that the two measurements agree to within their combined
uncertainties at ~25% (Hu et al., 2015). Meteorological conditions were monitored and recorded
during each balloon flight with a radiosonde (Met1, Grant Pass, OR, USA) attached to the tethered
line just below the balloon.
**2.4 BVOC emission rates**
2.4.1   Dynamic enclosure measurements
We used dynamic enclosure systems operated at low residence time to quantify vegetative BVOC
emissions following the procedure described by Ortega et al. (2008) and Ortega and Helmig
(2008). Two types of enclosures were used: branch and surface chambers. For branch enclosures,
a Tedlar® bag (Jensen Inert Products, Coral Springs, FL) was sealed around the trunk side of a
branch. For surface enclosures, the bag was placed around a circular Teflon® base (25 cm wide ×
16 cm height; see Fig. 2). For both branch and surface enclosures, the bag was connected to a
purge-air line and a sampling line, and positioned around the vegetation minimizing contact with
foliage. While purging the enclosure (see Section 2.4.3), the vegetation was allowed to acclimate
for 24 hours before BVOC sampling began. Samples were collected from the enclosure air,
concentrated onto solid-adsorbent cartridges (see Section 2.3) with an automated sampler, and
analyzed in-laboratory at the University of Colorado Boulder following the campaign (see S.I.
Section 1). Temperature and relative humidity were recorded inside and outside the enclosure (see
Fig. 2; S-THB-M002 sensors, Onset HOBO, Bourne, MA, USA) with a data logger (H21-USB,
Onset HOBO, Bourne, MA, USA). Additionally, photosynthetically active radiation (400-700 nm;
S-LIA-M003, Onset HOBO, Bourne, MA, USA) was measured inside the enclosure. Once
installed, enclosures were operated for 2-10 days. The tundra vegetation around TFS is
heterogeneous but most dominant species were sampled. Table 1 presents the median relative
percent cover of plant species in LTER experimental control plots at TFS (Gough, 2019) and
indicates whether plant species were present in surface or bag enclosures. The complete list of
species sampled and pictures of the enclosures are available in Figures S.I.3-S.I.15; the two
sampling sectors are highlighted in Fig.S.I.1. Surface enclosures were divided into three vegetation





types: *Salix* spp. (high isoprene emitter), *Betula* spp. (e.g., *Betula nana* dominance), and
miscellaneous (mix of different species, including lichens and mosses).

2.4.2   Emission rates

The emission rate (ER in µgC/m²/h) for surface enclosures was calculated as follows:
$ER_{surface} = \frac{(C_{out} - C_{in})Q}{S}$,                                          *(1)*
where $C_{in}$ and $C_{out}$ are the inlet and outlet analyte concentrations (in µgC/L), $Q$ is the purge air
flow rate (in L/h), and $S$ the surface area of the enclosure (in m²).
The ER for branch enclosures (in µgC/g/h) was calculated as follows:
$ER_{branch} = \frac{(C_{out} - C_{in})Q}{m_{dry}}$,                                          *(2)*
where $m_{dry}$ is the dried mass (in g) of leaves enclosed, determined by drying the leaves – harvested
after the experiment – at 60-70°C until a consistent weight was achieved (Ortega and Helmig,

2008).

2.4.3   Enclosure purge air

Purge air was provided by an upstream high-capacity oil-free pump providing positive pressure to
the enclosure, and equipped with an in-line $O_3$ scrubber to avoid loss of reactive BVOCs from
reaction with $O_3$ in the enclosure air and during sampling (Helmig, 1997; Pollmann et al., 2005).
The purge flow was set to 25 L/min and regularly checked using a volumetric flow meter (Mesa
Labs Bios DryCal Defender, Butler, NJ, USA). Excess air escaped from the open end (tied around
the Teflon® base) while the sample air flow was pulled into the sampling line (see below).

2.4.4   Sample collection

A continuous airflow of 400-500 mL/min was drawn from the enclosure through the sampling line.
A fraction of this flow was periodically collected at 265-275 mL/min on adsorbent cartridges (see
Section 2.3) using a 10-cartridge autosampler (Helmig et al., 2004). During sampling, cartridges
were kept at 40°C, *i.e.,* above ambient temperature, to prevent water accumulation on the adsorbent
bed (Karbiwnyk et al., 2002). Samples were periodically collected in series to verify lack of analyte
breakthrough. Time-integrated samples were collected for 120 min every 2 hours to establish





diurnal cycles of BVOC emission. Upon collection, samples were stored in the dark at ~4°C until
chemical analysis back at the University of Colorado Boulder.

### 2.4.5   Internal standards

In order to identify potential BVOC losses during transport, storage, and chemical analysis, 255
of the employed cartridges were pre-loaded with a four-compound standard mixture prior to the
field campaigns. These internal standard compounds (toluene, 1, 2, 3-trimethylbenzene, 1, 2, 3, 4-
tetrahydronaphtalene, and 1, 3, 5-triisopropylbenzene) were carefully chosen to span a wide range
of volatility ($C_7$-$C_{15}$) and to not interfere (*i.e.,* coelute) with targeted BVOCs. The recovery of these
four compounds was assessed at the end of the campaign, following the analytical procedure
described in S.I. Section 1. Recovery rates were 101.8 ± 13.5 % (toluene), 95.2 ± 20.1 % (1,2,3-
trimethylbenzene), 95.6 ± 26.6 % (1,2,3,4-tetrahydronaphtalene), and 100.9 ± 18.7 % (1,3,5-
triisopropylbenzene). These results indicate that, overall, BVOC losses during transport, storage,
and chemical analysis were negligible. Ortega et al. (2008) previously evaluated systematic losses
of analytes to enclosure systems similar to those used here. The same four-component standard
was introduced into the purge air flow of the enclosures to quantify losses as a function of
volatility. That work found median losses of MT and SQT on the order of 20-30%. The emission
rates presented here are therefore possibly biased low by a similar amount.

### 2.5 Peak fitting algorithm

The analysis of ambient air and enclosure chromatograms was performed using the TERN
(Thermal desorption aerosol GC ExploreR and iNtegration package) peak fitting tool implemented
in Igor Pro and available online at https://sites.google.com/site/terninigor/ (Isaacman-VanWertz et
al., 2017).

### 2.6 Ancillary parameters

*Meteorological parameters.* A suite of meteorological instruments was deployed on the 4 m tower.
Wind speed and direction were measured at ~4 m above ground level with a Met One 034B-L
sensor. As described by Van Dam et al. (2013), temperature was measured at three different heights
using RTD temperature probes (model 41342, R.M. Young Company, Traverse City, MI) housed
in aspirated radiation shields (model 43502, R.M. Young Company, Traverse City, MI). Regular
same-height inter-comparisons were conducted to test for instrumental offsets. Incoming and



reflected solar radiation were recorded with LI200X pyranometers (Campbell Scientific
Instruments).
In addition, historical (1988-2019) meteorological data recorded by TFS Environmental Data
Center are available at: https://toolik.alaska.edu/edc/abiotic_monitoring/data_query.php
*Particle measurements.* A Met One Instruments Model 212-2 8-channel (0.3 to 10 µm) particle
profiler was operated continuously on the roof of the weatherproof instrument shelter. This
instrument uses a laser-diode based optical sensor and light scatter technology to detect, size, and
count particles (http://mail.metone.com/particulate-Aero212.htm).
*Nitrogen oxides.* Nitrogen oxides ($NO_x$) were measured with a custom-built, high sensitivity (~5
pptv detection limit) single-channel chemiluminescence analyzer (Fontijn et al., 1970). The
instrument monitors nitric oxide (NO) and nitrogen dioxide ($NO_2$) in ambient air using a photolytic
converter. Automated switching valves alternated between NO and $NO_2$ mode every 30 minutes.
Calibration was accomplished by dynamic dilution of a 1.5 ppm compressed NO gas standard
(Scott-Marrin, Riverside, CA, USA).

**2.7 Theoretical response of isoprene emissions to temperature in MEGAN2.1**

We applied our isoprene emission measurements to evaluate the temperature response algorithms
embedded in MEGAN2.1 (Guenther et al., 2012). Theoretical isoprene emission rates ($F_T$) were
calculated for TFS as:
$F_T = C_{CE}\, \gamma_T \, \sum_j \varkappa_j\, \varepsilon_j$                                    (3)
where $C_{CE}$ is the canopy environment coefficient (assigned a value that results in $\gamma_T = 1$ under
standard conditions), and $\varepsilon_j$ is the emission factor under standard conditions for vegetation type $j$
with fractional grid box areal coverage $\varkappa_j$. We used $\sum_j \varkappa_j \varepsilon_j = 2766$ µg/m²/h at TFS based on the
high    resolution    (1    km)    global    emission    factor    input    file    available    at
https://bai.ess.uci.edu/megan/data-and-code/megan21. The  temperature activity factor ($\gamma_T$) was
calculated as:
$\gamma_T = E_{opt} \times \dfrac{200\, e^{95\, x}}{200 - 95 \times (1 - e^{200\, x})}$                    (4)
with



$$x = \frac{\frac{1}{T_{opt}} - \frac{1}{T}}{0.00831} \qquad (5)$$
$$E_{opt} = 2 \times e^{0.08(T_{10} - 297)} \qquad (6)$$
$$T_{opt} = 313 + 0.6(T_{10} - 297), \qquad (7)$$
where $T$ is the enclosure ambient air temperature and $T_{10}$ the average enclosure air temperature
over the past 10 days.

## 3. Results and Discussion

### 3.1 Ambient air mixing ratios

#### 3.1.1 Isoprene and oxidation products

Figure 3 (top panels) shows the time-series of isoprene mixing ratios in ambient air recorded over
the course of this study at TFS with the GC system. Mixing ratios were highly variable and ranged
from below the quantification limit to 505 pptv (mean of 36.1 pptv). The PTR-ToF-MS gave
similar results (see Fig.S.I.16a). These mixing ratios fall within the range of values reported in the
Eurasian taiga (e.g., Hakola et al., 2000, 2003; Lappalainen et al., 2009). For example, Hakola et
al. (2003) reported a maximum monthly mean mixing ratio of 98 pptv (in July) in Central Finland
while Hakola et al. (2000) observed mixing ratios ranging from a few pptv to ~600 pptv in Eastern
Finland. In general, however, BVOC emissions in the Eurasian taiga are relatively low compared
to forest ecosystems in warmer climates and are dominated by monoterpenes (Rinne et al., 2009).
Isoprene mixing ratios peaked on August 1, 2018 around 4 pm and on June 20, 2019 around 10
pm, respectively. These two peaks occurred 3-5 hours after the daily maximum ambient
temperature was reached (17.8°C in 2019 and 21.8°C in 2019 – see Fig. 3). The isoprene peak on
June 20, 2019 was concomittant with enhanced acetonitrile mixing ratios and particle counts (see
Fig. 4), reflecting unusually hazy conditions that day at TFS. We attribute the particle and
acetonitrile enhancements to intense wildfires occurring across the Arctic Circle at that time – most
of them in southern Alaska and Siberia (Earth Observatory, 2019). Acetonitrile increased by a
factor of 4 during this event, compared to a factor of 21 increase for isoprene. The higher emission
factor for acetonitrile *vs.* isoprene from biomass burning in boreal forests (Akagi et al., 2011) and
the relatively short lifetime of isoprene (Atkinson, 2000) indicate that the observed isoprene
enhancement was due to fresh local biogenic emissions rather than transported wildfire emissions.



Over the course of this study, we recorded MACR and MVK mixing ratios respectively ranging
from below the quantification limit to 95 pptv (12.4 ± 16.1 pptv; mean ± standard deviation) and
from below the quantification limit to 450 pptv (43.1 ± 66.7 pptv; see Fig. 3, top panels). The PTR-
ToF-MS gave similar results (see Fig.S.I.16b). Median NO and $NO_2$ mixing ratios of 21 and 74
pptv, respectively, during the 2019 campaign (not shown) suggest a low-$NO_x$ environment, in line
with previous studies at several Arctic locations (Bakwin et al., 1992; Honrath and Jaffe, 1992).
Under such conditions, MACR and MVK mixing ratios should be used as upper estimates as it has
been noted that some low-$NO_x$ isoprene oxidation products (isoprene hydroxyhydroperoxides) can
undergo rearrangement in GC and PTR-MS instruments and be misidentified as MACR and MVK
(Rivera-Rios et al., 2014). We found a high correlation between MACR and MVK ($R^2 = 0.95$, p <
0.01) and between these two compounds and isoprene ($R^2 \sim 0.80$, p < 0.01). Increases of MACR
and MVK mixing ratios above the background were mostly concomitant with isoprene increases,
suggesting that atmospheric or within-plant oxidation of isoprene was their main source
(Biesenthal et al., 1997; Hakola et al., 2003; Jardine et al., 2012). The mean ratio of MVK to
MACR was 2.7, within the range reported by earlier studies (e.g., Apel et al., 2002; Biesenthal and
Shepson, 1997; Hakola et al., 2003; Helmig et al., 1998), and no clear diurnal cycle in the ratio
was found.

3.1.2    Isoprene vertical profiles

Figure 5 shows vertical profiles (0 to ~250 m a.g.l.) of isoprene mixing ratios derived from the 30-
min tethered balloon samples collected on June 15 and 16, 2019 (see Section 2.3). Temperature
profiles (see Fig.S.I.17) indicate that most of the flights were performed in a convective boundary
layer (Holton and Hakim, 2013). A nocturnal boundary layer was, however, observed in the first
~50 m from ~2 am to ~4:30 am (see Fig.S.I.17e-f) – with temperature increasing with elevation.
Except during the last flight, isoprene mixing ratios were in the range of background levels (~0-
50 pptv) reported with the GC-MS (see Section 3.1.1). Samples collected on June 16, 2019 from
4 to 4:30 am show decreasing isoprene mixing ratios with increasing elevation, suggesting higher
levels (25-50 pptv) in the nocturnal boundary layer than above. Samples collected from 10-10:30
am on June 16 (*i.e.,* during the last flight) showed a pronounced gradient, with 200 pptv at ground
level and decreasing mixing ratios with elevation. This maximum at ground-level is consistent
with a surface source (Helmig et al., 1998) and can likely be attributed to a temperature-driven





increase of isoprene emissions by the surrounding vegetation. Indeed, the ambient temperature at
ground-level was higher during that flight than during the previous ones (see Fig.S.I.17h).
Interestingly, the GC-MS and the PTR-ToF-MS did not capture this 200 pptv maximum (see Fig.
3 and Fig.S.I.16), which may be because the balloon flights were performed at a different location
(near sampling sector B, see Fig.S.I.1) surrounded by a higher fraction of isoprene-emitting shrubs
(willow).

### 3.1.3 Monoterpenes and Sesquiterpenes

MT mixing ratios ranged from 3 to 537 pptv (14 ± 18 pptv; median ± standard deviation) during
the 2019 campaign according to the PTR-ToF-MS measurements. Using the GC-MS/FID, we were
able to detect and quantify the following MT in ambient air: α-pinene, camphene, sabinene, p-
cymene, and limonene. Mean mixing ratios are reported in Table 2 (for values lower than the LOQ,
mixing ratios equal to half of the LOQ are used). These compounds have been previously identified
as emissions of the widespread circumpolar dwarf birch *Betula nana* (Li et al., 2019; Vedel-
Petersen et al., 2015) and other high Arctic vegetation (Schollert et al., 2014). The quantification
frequency of camphene, sabinene, p-cymene, and limonene was low (see Table 2) and MT mixing
ratios in ambient air were dominated by α-pinene. Several prior studies performed at boreal sites
have similarly identified α-pinene as the most abundant monoterpene throughout the growing
season (e.g., Hakola et al., 2000; Lindfors et al., 2000; Spirig et al., 2004; Tarvainen et al., 2007).
We did not detect any sesquiterpene in ambient air above the 2 pptv instrumental LOQ.
Overall, isoprene and α-pinene dominated the ambient air BVOC profile at TFS, respectively
constituting ~72% and ~24% of total BVOCs quantified in ambient air (on a mixing-ratio basis).

### 3.2 Emission rates

#### 3.2.1 Branch enclosures

A branch enclosure experiment was performed from July 27 to August 2, 2018 on *Salix glauca* to
investigate BVOC emission rates per dry weight plant biomass (see Fig.S.I.5). Isoprene emission
rates ranged from <0.01 to 11 µgC/g/h, in line with results reported for the same species at
Kobbefjord, Greenland (0.8 to 12.1 µgC/g/h) by Vedel-Petersen et al. (2015) and Kramshøj et al.
(2016; Supplementary Table 5). The quantified MTs had emissions averaging two orders of
magnitude lower than those of isoprene (0.01 vs 1 µgC/g/h). Emission rates for the sum of α-



pinene, β-pinene, limonene, camphene, and 1,8-cineole ranged from <0.01 to 0.06 µgC/g/h. These
results are again in good agreement with those reported for the same species at Kobbefjord (~0.01
µgC/g/h) by Kramshøj et al. (2016; Supplementary Table 5).

3.2.2   Surface emission rates

The isoprene surface emission rate, as inferred from surface enclosures, was highly variable and
ranged from 0.2 to ~2250 µgC/m$^2$/h (see Fig. 6). The 2250 µgC/m$^2$/h maximum, reached on June
26, 2019, is higher than maximum values reported at TFS by Potosnak et al. (2013) (1200
µgC/m$^2$/h) and in a high-latitude (58°N) *Salix* plantation by Olofsson et al. (2005) (730 µgC/m$^2$/h).
It should be noted that these experiments were likely performed at different ambient temperatures.
We further investigate the temperature dependency of isoprene emissions in Section 3.3. Elevated
surface emission rates (*i.e.,* > 500 µgC/m$^2$/h) were all observed while sampling enclosures
dominated by *Salix* spp.. At TFS, the overall mean isoprene emission rate amounted to 85
µgC/m$^2$/h while the daytime (10 am-8 pm) and midday (11 am-2 pm) means were 140 and 213
µgC/m$^2$/h, respectively. To put this in perspective, isoprene surface emission rates were much
lower than for mid-latitude or tropical forests. For example, average midday fluxes of 3000
µgC/m$^2$/h were reported in a northern hardwood forest in Michigan (Pressley et al., 2005), while
several reports of isoprene emissions from tropical ecosystems give daily estimates of 2500-3000
µgC/m$^2$/h (Helmig et al., 1998; Karl et al., 2004; Rinne et al., 2002).
Figure 7 shows the measured surface emission rates for α-pinene, β-pinene, limonene, and 1,8-
cineole. While p-cymene, sabinene, 3-carene, and isocaryophyllene (SQT) were detected in some
of the surface enclosure samples, we focus the discussion on the most frequently quantified
compounds. Regardless of the species, emission rates remained on average below 1 µgC/m$^2$/h over
the course of the study (see Table 3). These results are at the low end of emission rates reported
for four vegetation types in high Arctic Greenland (Schollert et al., 2014), but in line with results
reported at Kobbefjord, Greenland by Kramshøj et al. (2016; Supplementary Table 4).
Figures 8a-c show the mean diurnal cycle (over the two campaigns) of isoprene surface emission
rates for different vegetation types (see Fig.S.I.3-15 for nomenclature). Regardless of the
vegetation type, isoprene emission rates exhibited a significant diurnal cycle with an early
afternoon maximum, in line with the mean diurnal cycle of enclosure temperature and PAR. These





results are in line with the well-established diurnal variation of BVOC emissions in environments
ranging from Mediterranean to boreal forests (e.g., Fares et al., 2013; Liu et al., 2004; Ruuskanen
et al., 2005; Zini et al., 2001). Despite the relatively low MT emission rates, a significant diurnal
cycle was also observed with peak total MT emissions of ~1 µgC/m²/h during early afternoon for
both *Salix* spp. and *Betula* spp. (Fig. 8e-f). A summary of emission rates per vegetation type and
time of day is given in Table 3. It should be noted that the two field campaigns were carried out
during the midnight sun period, which could possibly sustain BVOC emissions during nighttime.
As can be seen in Table 3 and Fig. 8, PAR and BVOC emissions significantly decreased at night
but were still detectable. These results confirm those obtained by Lindwall et al. (2015) during a
24-hour experiment with five different Arctic vegetation communities.
The ratio of total MT (given by the sum of α-pinene, β-pinene, limonene, and 1,8-cineole)
emissions to isoprene emissions was an order of magnitude higher for *Betula* spp. (0.22) than for
*Salix* spp. (0.03). This result, driven by the relatively lower isoprene emissions of *Betula* spp., is
in line with earlier studies, suggesting similar emission characteristics for Arctic plants (e.g.,
Kramshøj et al., 2016; Vedel-Petersen et al., 2015).

**3.3 Response of isoprene emissions to temperature**

The Arctic has warmed significantly during the last three decades and temperatures are projected
to increase an additional 5-13°C by the end of the century (Overland et al., 2014). Heat wave
frequency is also increasing in the terrestrial Arctic (Dobricic et al., 2020). For example, western
Siberia experienced an unusually warm May in 2020, with temperatures of 20-25°C (Freedman
and Cappucci, 2020). In that context, numerous studies have pointed out the likelihood of increased
BVOC emissions due to Arctic warming and associated vegetation and land cover change (Faubert
et al., 2010; Potosnak et al., 2013; Rinnan et al., 2011; Tiiva et al., 2008).
Over the course of the two field campaigns at TFS, BVOC surface emission rates were measured
over a large span of enclosure temperatures (2-41°C). While MT emissions remained low and close
to the detection limit thus preventing robust quantification of any emission-temperature
relationship, isoprene emissions significantly increased with temperature (Fig.9). Figure 9
combines isoprene emission rates from different surface enclosures, with results normalized to
account for differing leaf area and species distributions (with *Salix* spp. the dominant emitter).
Specifically, we divided all fluxes by the enclosure-specific mean emission at 20 ± 1°C. Emission
rates are often standardized to 30°C but we employ 20°C here owing to the colder growth
environment at TFS (Ekberg et al., 2009). The isoprene emission-temperature relationship
observed at TFS (in blue) is very similar to that reported by Tang et al. (2016) at Abisko (Sweden;
in pink) for tundra heath (dominated by evergreen and deciduous dwarf shrubs). Results at TFS
and Abisko both point to a high isoprene-temperature response for Arctic ecosystems (Tang et al.,
2016). This is further supported by two warming experiments performed in mesic tundra heath
(dominated by *Betula nana*, *Empetrum nigrum*, *Empetrum hermaphroditum*, and *Cassiope*
*tetragona*) and dry dwarf-shrub tundra (co-dominated by *Empetrum hermaphroditum* and *Salix*
*glauca*) in Western Greenland (Kramshøj et al., 2016; Lindwall et al., 2016). Kramshøj et al.
(2016) observed a 240% isoprene emission increase with 3°C warming, while Lindwall et al.
(2016) reported a 280% increase with 4°C warming. The observationally-derived emission-
temperature relationship derived here for TFS reveals a 180-215% emission increase with 3-4°C
warming, adding to a growing body of evidence indicating a high isoprene-temperature response
in Arctic ecosystems.
The MEGAN2.1 modeling framework is commonly used to estimate BVOC fluxes between
terrestrial ecosystems and the atmosphere (e.g., Millet et al., 2018). Here, we apply the TFS
observations to evaluate the MEGAN2.1 emission-temperature relationship for this Arctic
environment. Figure 9 shows that the model temperature algorithm provides a close fit with
observations below 30°C, with a 170-240% emission increase for a 3-4°C warming. However, the
model predicts a leveling-off of emissions at approximately 30-35°C, whereas our observations
reveal no such leveling-off within the 0-40°C enclosure temperature range (Fig. 9). Current models
can therefore be expected to strongly underpredict isoprene emissions in this region for
temperatures above 30°C.
To put the above finding in perspective, the highest air temperature on record at TFS (1988-2019)
is 26.5°C, and the mean summertime (June-August) temperature over that period is 9°C. Only 1-
23 and 0-4 days per year over that timespan recorded daily maximum temperatures above 20°C
and 25°C, respectively. If global greenhouse gas emissions continue to increase, temperatures are
expected to rise 6-7°C in northern Alaska by the end of the century (annual average; Markon et
al., 2012) while the number of days with temperatures above 25°C could triple (Lader et al., 2017).
Based on current climate conditions and this rate of change, the MEGAN2.1 algorithm can still be



expected to adequately represent the temperature dependence response of Arctic ecosystems for
the near and intermediate-term future.
**4. Implications and conclusions**
While BVOC ambient concentrations and emission rates have been frequently measured in boreal
ecosystems, Arctic tundra environments are scarcely studied. We provide here summertime BVOC
ambient air mixing ratios and emission rates at Toolik Field Station, on the north flanks of the
Brooks Range in northern Alaska. These data provide a baseline to investigate future changes in
the BVOC emission potential of the Arctic tundra environment. Elevated isoprene surface
emission rates (> 500 µgC/m$^2$/h) were observed for *Salix* spp., a known isoprene emitter. The
reponse to temperature of isoprene emissions in enclosures dominated by *Salix* spp. increased
exponentially in the 0-40°C range, likely conferring greater thermal protection for these plants.
Our study indicates that the temperature algorithm in MEGAN2.1, a widely used modelling
framework for BVOC emissions, provides a good fit with observations in the Arctic tundra for
temperatures below 30°C. However, more studies are needed to better constrain the warming
response of isoprene and other BVOCs for a wider range of Arctic species.
In the context of a widespread increase in shrub abundance (including *Salix* spp.) in the Arctic
(Berner et al., 2018; Sturm et al., 2001) due to a longer growing season and enhanced nutrient
availability, our results support earlier studies (e.g., Valolahti et al., 2015) suggesting that climate-
induced changes in the Arctic vegetation composition will likely significantly affect the BVOC
emission potential of the Arctic tundra. As discussed extensively by Peñuelas and Staudt (2010)
and Loreto and Schnitlzer (2010), emissions of BVOCs might be largely beneficial for plants,
conferring them higher protection from abiotic stressors (*e.g.,* heat, air pollution, high irradiance)
which are predicted to be more severe in the future. Arctic warming may thus favor BVOC-
emitting species even further.
**Data availability**
Data are available upon request to the corresponding author.
**Author contribution**



DH, LH, and DBM designed the experiments and acquired funding. HA led the two field
campaigns with significant on-site contribution from KM, JH, LH, DBM, KC, JM, CW, TM, and
DH. JH designed and built most of the instruments used in this study. CW acquired the PTR-ToF-
MS data during the second campaign and DK performed data analysis. MSBH identified the plant
species and provided guidance during the field campaigns. KM and HA analyzed the samples in
the lab. HA analyzed all the data and prepared the manuscript with contributions from all co-
authors.
**Competing interests**
The authors declare no competing interests.
**Acknowledgements**
The authors would like to thank CH2MHill Polar Services for logistical support, the Toolik Field
Station staff for assistance with the measurements, and Ilann Bourgeois and Georgios Gkatzelis
for helpful discussions. The authors also appreciate the help of Anssi Liikanen who offered kind
assistance collecting BVOC samples with the tethered balloon and Wade Permar who helped with
PTR-ToF-MS measurements. Finally, the authors gratefully acknowledge Claudia Czimczik and
Shawn Pedron at the University of California Irvine for letting us use their soil chamber collars.
This research was funded by the National Science Foundation grant #1707569. Undergraduate
students Katelyn McErlean, Jacob Moss, and Kaixin Cui received financial support from the
University of Colorado Boulder's Undergraduate Research Opportunities Program (UROP;
reference #5352323, #4422751, and #4332562, respectively).





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





Table 1: Year 2017 median relative percent cover of plant species in moist acidic tundra long-term
ecological research (LTER) experimental control plots at Toolik Field Station. The last column
indicates whether plant species were present in surface or bag enclosure experiments in this study.

| Plant name | Relative land surface cover in moist acidic tundra (%) (Gough, 2019) | Present in surface or bag enclosures |
|---|---|---|
| *Andromeda polifolia* | 0.6 | yes |
| *Betula nana* | 14.4 | yes |
| *Carex bigelowii* | 1.0 | yes |
| *Cassiope tetragona* | 2.0 | yes |
| *Empetrum nigrum* | 3.8 | yes |
| *Eriophorum vaginatum* | 8.6 | yes |
| *Ledum palustre* | 10.5 | yes |
| *Mixed Lichens* | 2.1 | yes |
| *Mixed moss* | 6.0 | yes |
| *Pedicularis lapponica* | 0.6 | no |
| *Polygonum bistorta* | 0.6 | no |
| *Rubus chamaemorus* | 20.2 | no |
| *Salix pulchra* | 4.9 | yes |
| *Vaccinium uliginosum* | 1.9 | yes |
| *Vaccinium vitis-idaea* | 6.6 | yes |










Table 2: Average mixings ratios with standard deviation, along with minimum (min) and
maximum (max) values and quantification frequency (QF) of the measured monoterpenes in
ambient air. LOQ stands for limit of quantification. For values lower than the LOQ, mixing ratios
equal to half of the LOQ were used to calculate the mean.

|  | mean ± standard deviation (pptv) | Min (pptv) | Max (pptv) | QF (%) |
|---|---|---|---|---|
| α-pinene | 11.7 ± 8.1 | < LOQ | 61.6 | 88 |
| camphene | < LOQ | < LOQ | 21.9 | 11 |
| sabinene | < LOQ | < LOQ | 34.2 | 11 |
| p-cymene | 2.0 ± 1.9 | < LOQ | 12.3 | 32 |
| limonene | < LOQ | < LOQ | 2.9 | < 1 |

















Table 3: Isoprene and monoterpenes (sum of α-pinene, β-pinene, limonene, and 1,8-cineole) surface
emission rates per vegetation type. Miscellaneous refers to a mix of different species, including lichens
and moss tundra (see Fig.S.I.3-15). Daytime refers to 10 am-8 pm, midday to 11 am-2 pm, and nighttime
to 11 pm-5 am (Alaska Standard Time).

| | mean ± standard deviation ($\mu gC/m^2/h$) | daytime mean ± standard deviation ($\mu gC/m^2/h$) | midday mean ± standard deviation ($\mu gC/m^2/h$) | nighttime mean ± standard deviation ($\mu gC/m^2/h$) |
|---|---|---|---|---|
| isoprene | | | | |
| *Salix* spp. | 149 ± 327 | 232 ± 400 | 334 ± 473 | 7 ± 10 |
| *Betula* spp. | 12 ± 30 | 19 ± 38 | 28 ± 37 | 5 ± 14 |
| Miscellaneous | 38 ± 81 | 57 ± 100 | 104 ± 135 | 21 ± 64 |
| monoterpenes | | | | |
| *Salix* spp. | 0.8 ± 1.3 | 1.1 ± 1.5 | 1.4 ± 1.7 | 0.4 ± 1.0 |
| *Betula* spp. | 0.5 ± 0.6 | 0.7 ± 0.7 | 1.0 ± 0.8 | 0.2 ± 0.2 |
| Miscellaneous | 1.1 ± 1.4 | 1.3 ± 1.6 | 1.7 ± 2.0 | 1.0 ± 1.4 |












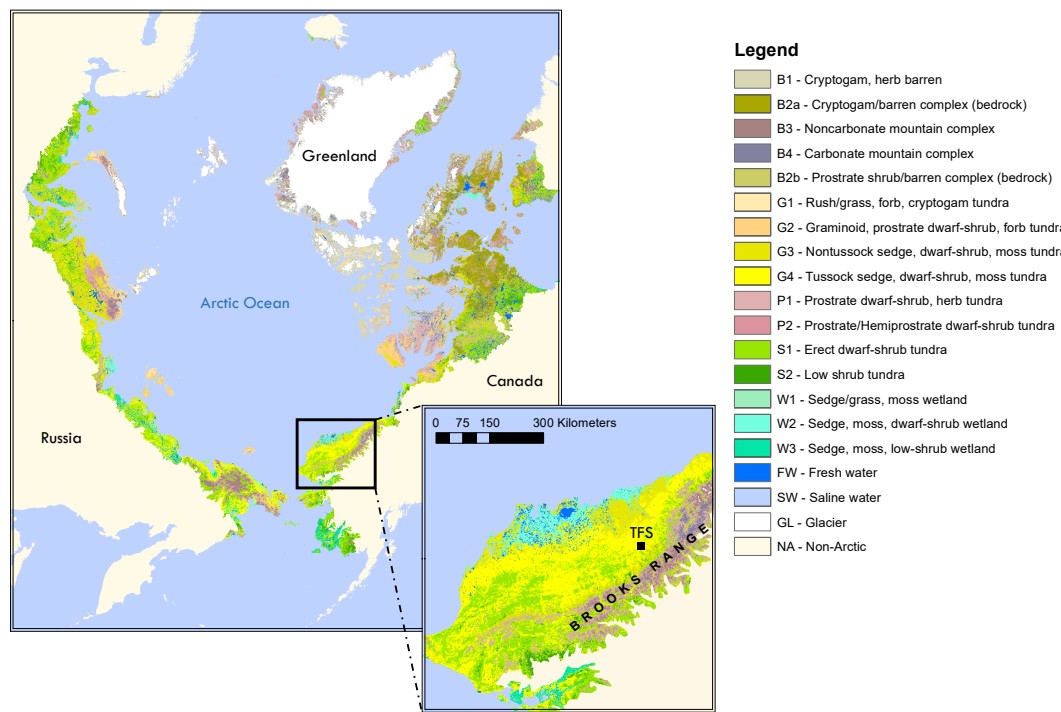


Figure 1: Location of Toolik Field Station (TFS) on the north flanks of the Brooks Range in northern Alaska

along with arctic vegetation type. This Figure was made using the raster version of the Circumpolar Arctic

Vegetation Map prepared by Raynolds et al. (2019) and publicly available at www.geobotany.uaf.edu.













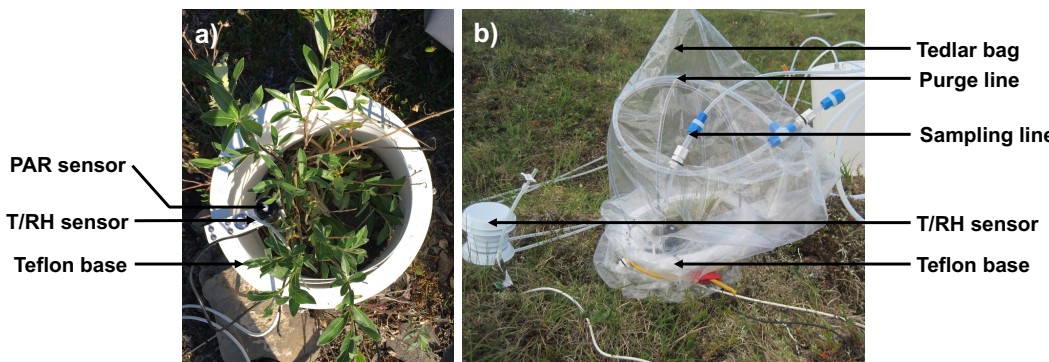


Figure 2: Photographs of a surface enclosure experiment setup at Toolik Field Station, Alaska. a) The first step of the installation consisted in positioning the Teflon® base around the vegetation of interest along with temperature (T), relative humidity (RH), and photosynthetically active radiation (PAR) sensors. b) The second step consisted in positioning the Tedlar® bag around the base. The bag was connected to a purge air and a sampling line. An additional T/RH sensor was also positioned outside the bag.















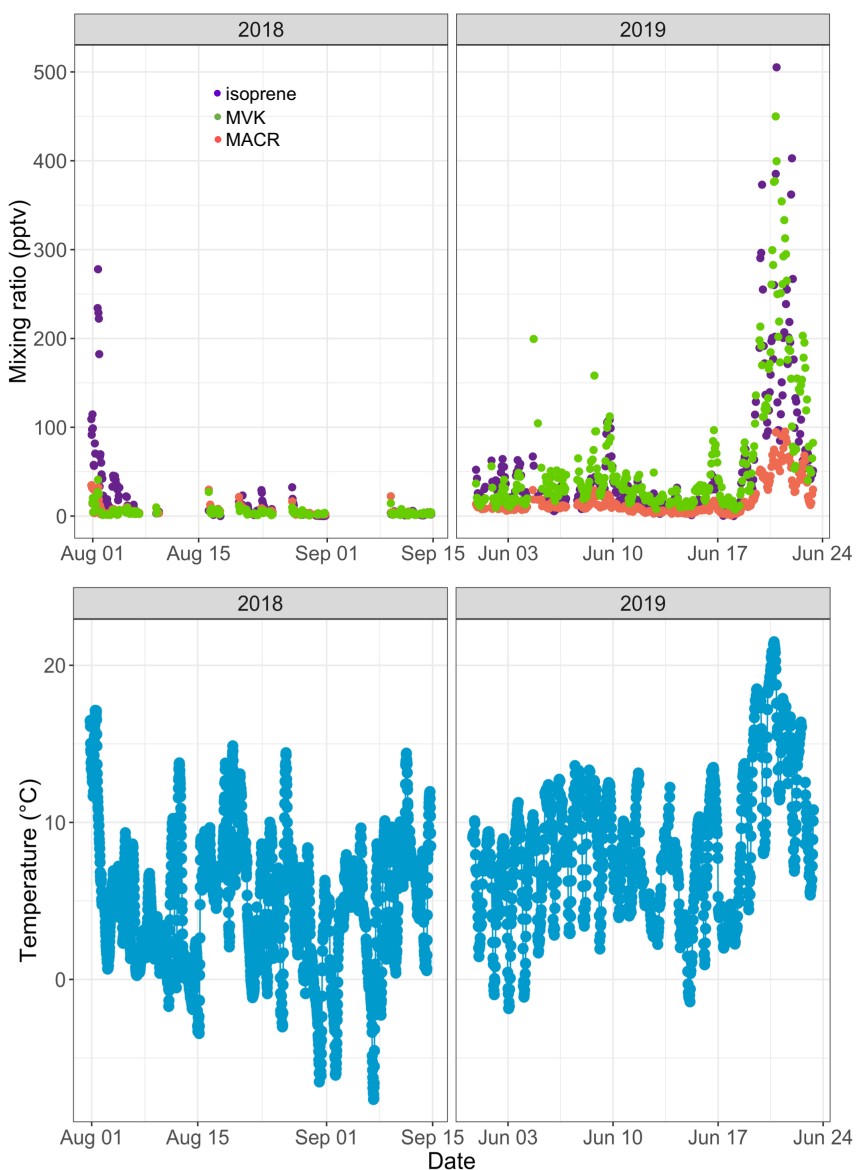


Figure 3: Time-series of isoprene (purple), methylvinylketone (MVK, green), and methacrolein (MACR, salmon) mixing ratios (in pptv) in ambient air at Toolik Field station (top panels) and of 30-min-averaged ambient temperature (in °C) at 4 meters above ground level (bottom panels).










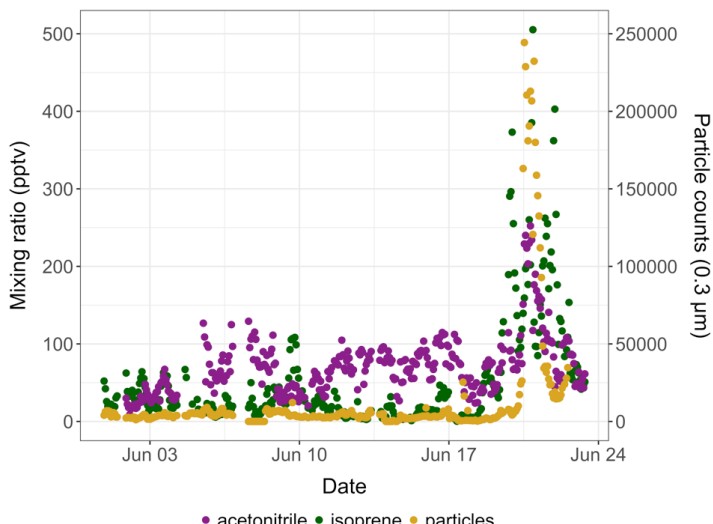


Figure 4: Time-series of isoprene (green) and acetonitrile (purple) mixing ratios (in pptv) and of 0.3 μm
particle counts (yellow) in ambient air at Toolik Field station in June 2019.








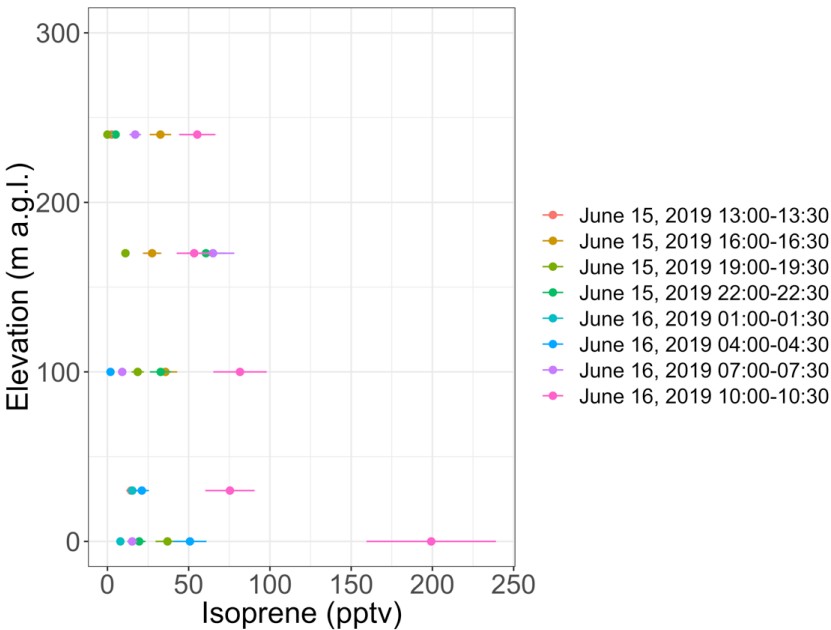


Figure 5: Vertical profile of isoprene mixing ratios as inferred from 30-min samples collected with a
tethered balloon. The error bars show the analytical uncertainty for isoprene (20 %). Samples with an
isoprene mixing ratio lower than blanks were discarded. Hours are in Alaska Standard Time (UTC-9).














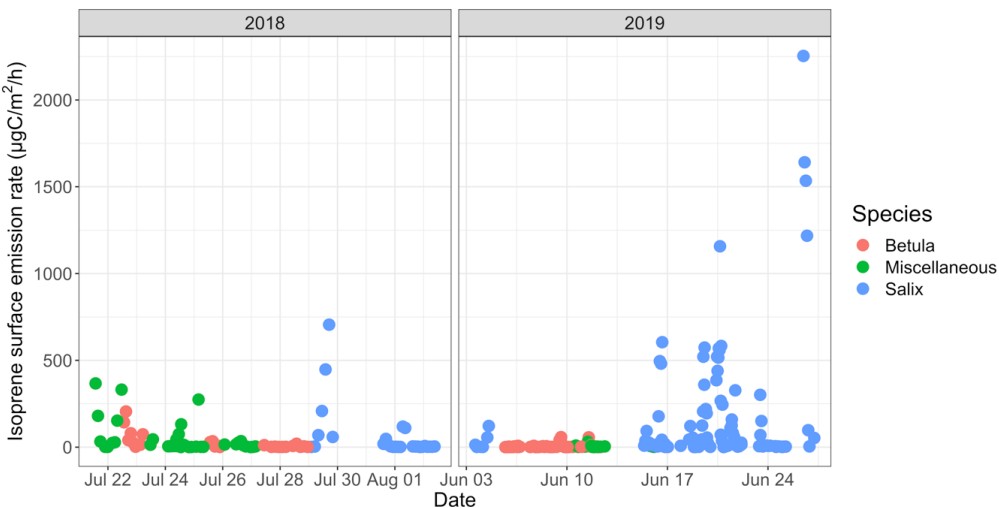


Figure 6: Time-series of isoprene surface emission rates (in µgC/m$^2$/h) for different vegetation types.











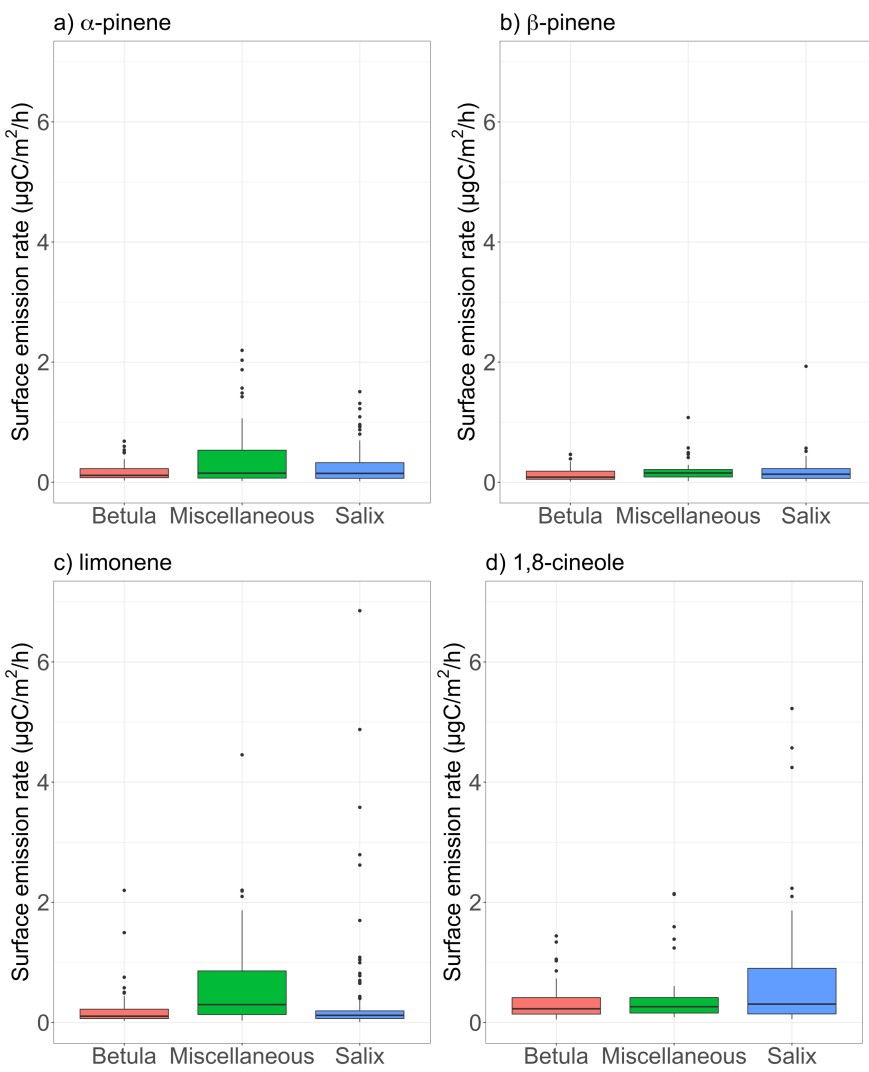


Figure 7: Surface emission rates of various monoterpenes (in µgC/m²/h) for different vegetation types. The lower and upper hinges correspond to the first and third quartiles. The upper (lower) whisker extends from the hinge to the largest (smallest) value no further than $1.5 \times IQR$ from the hinge, where $IQR$ is the inter-quartile range (i.e., the distance between the first and third quartiles). The notches extend $1.58 \times IQR/\sqrt{n}$ and give a ~95% confidence interval for medians.

953


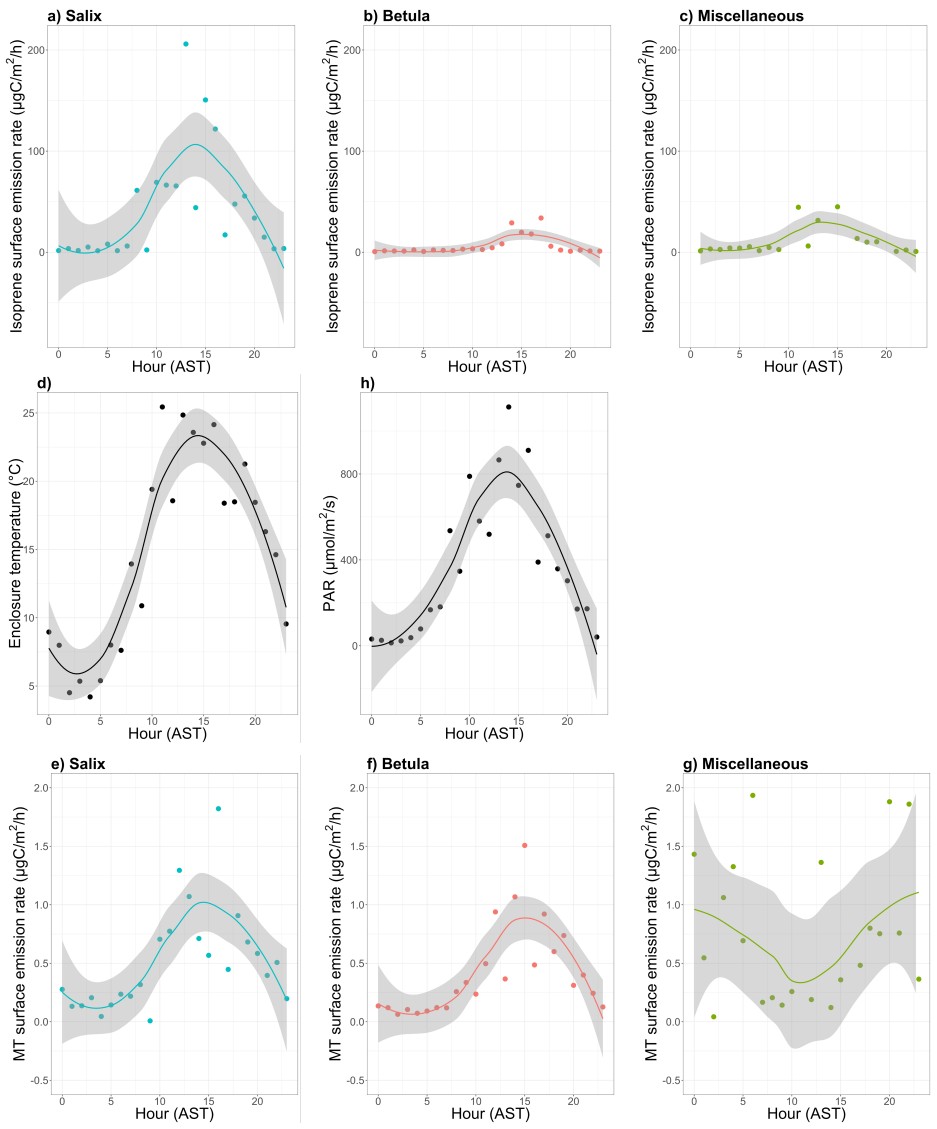

954

Figure 8: Mean diurnal cycle of isoprene (a-c) and monoterpenes (MT; e-g) surface emission rates (in
$\mu gC/m^2/h$ – note the difference scale on the y-axis), d) enclosure temperature (in °C), and h) enclosure
photosynthetically active radiation (PAR in $\mu mol/m^2/s$). The dots represent the hourly means. The line is
the smoothed conditional mean while the grey shaded region indicates the 95% confidence interval. Hours
are in Alaska Standard Time (UTC-9) and correspond to the end of the 2-hr sampling period for isoprene
and MT emission rates. MT corresponds here to the sum of α-pinene, β-pinene, limonene, and 1,8-cineole.

961

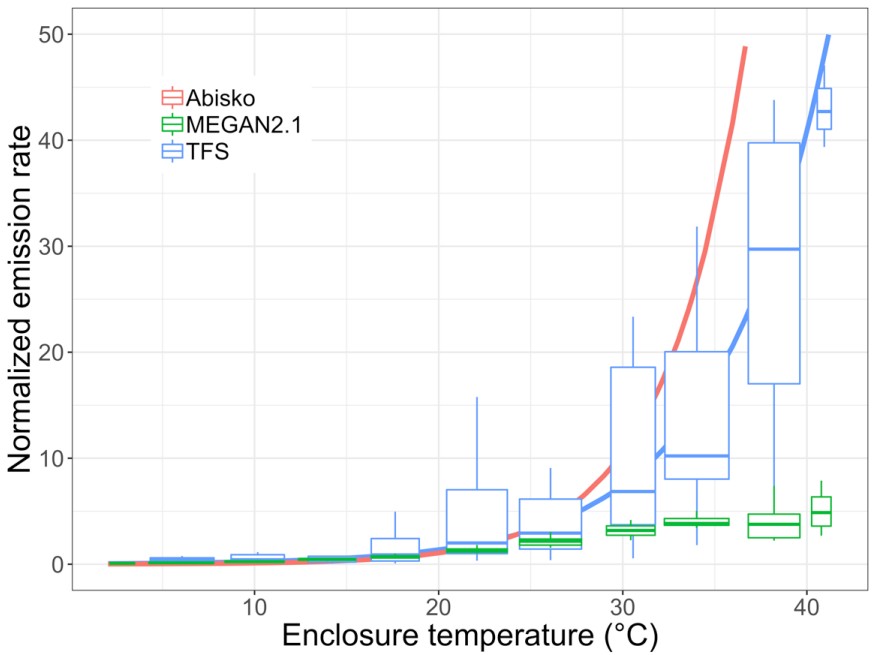

962

Figure 9: Normalized isoprene surface emission rate (emissions at 20°C set equal to 1.0) as a function of
enclosure temperature (in °C). This figure shows the response to temperature as observed at Toolik Field
Station (TFS, in blue) and Abisko, Sweden (in pink; Tang et al., 2016), and as parameterized in MEGAN2.1
(in green). The blue solid line is the exponential fit at TFS. It should be noted that the enclosure temperature
was on average 5-6°C warmer than ambient air due to greenhouse heating.