# Peer review of "Biogenic volatile organic compound ambient mixing ratios and emission rates"

_Biogeosciences, 2020_

## Referee Comment (RC1) · Anonymous Referee #1 · 24 Aug 2020

This paper aims at quantifying terpenoid mixing ratios and emission rates of dominant vegetation in northern Alaska. The authors have intensively compared their data with the published data from northern Sweden and Greenland and derived site-specific temperature response curve. This paper is well written and the data from this paper can provide base quantification of BVOC emissions from this less-studied area. A variety of measurements have been used in this paper and I, as a modeler, will leave the measurement part to other reviewer(s).

My main concern of this paper is that the mixing ratio measurements are not much linked to the emission rate measurements. It is a lot of data presented (which was

good), but I think the authors should bring these data together to present a whole story. Then, another part is about comparing emission ranges with literature values. The measurement conditions could vary largely and also in different periods of growing season. It is difficult to directly conclude that the measurement values are in the range of published values. I think standardized emission rates (using commonly-used Guenther algorithm) are needed in this case.

Here are some detailed comments:

Introduction: I would think one to two sentences could be needed to justify the importance of studying BVOC emissions on impacting atmospheric chemistry from this less-polluted arctic region . Then I think the aim of this study should be elevated, so what are the main aim of this study apart from quantifying emissions and mixing ratios.

L52-53, the field warming increases of BVOC emission is not only seen with long-term warming but also found with a short-term field warming like 3 years in the same area.

L97, please describe the start and end of a normal growing season for this site.

Table1 Rhododendron tomentosum seems the 2nd highest covered in this area and why not present emission from this species separately?

Fig. 3, I have a bit difficulty to find all measurements points at different heights. Suggest to use more distinguishing colors combining with different symbols.

L362, as far as I know, Cassiope tetragona is also a MT emitter.

L372-L380, how valid it is to state that the values are in the range of published values if focusing on the emission rates potentially measured at very different temperature and light conditions. I would suggest comparing the standardized emission rates with other studies if possible.

L435, What does this mean "account for differing leaf area. . ."? This is the emission rate of the per ground area, right? Please clarify.

L436, If dividing all fluxes with the standard emission rates at 20 degree, then it gives a multiplication of environmental responses (unit-less). As PAR is measured in the chamber, why not take away the light variation part before only looking at temperature response curve? Then about Fig. 9, how did you deal with the MEGAN temperature response curve, as I did not see the normalized emission rate to 1 around 30 degree?

L464-L468, MEGAN uses leaf temperature, not ambient air temperature for emission estimations. With predicted strong increase of air temperature in the Arctic, it still remain largely unknown (interesting to know) how plant leaf temperature can change and thus impact on BVOC emissions. I think it is important to have this in the discussion context.

---

## Referee Comment (RC2) · Anonymous Referee #2 · 22 Sep 2020

Major comments The manuscript presents results from a series of well-designed experiments that explore emissions of isoprene and monoterpenes from Artic plants to the atmosphere. The experimental design is sound and well-described, except for a few minor points noted below. The main conclusion is that the rapidly warming Arctic will cause sharply increased emissions of isoprene, which previous studies have shown to have a significant impact on atmospheric chemistry. Overall, this paper supports previous research findings, but the detail and atmospheric chemistry perspective make it a valuable contribution to the literature.

While the focus of the paper is from the atmosphere-exchange perspective, several

ecophysiological concepts which have been discussed previously for Arctic BVOC emissions should be addressed. First, given the emphasis placed in the paper on the response of emissions to warming, the acclimation process should be addressed (see one reference below). Second, temperature in the current study refers to air temperature, but isoprene and other MT emissions respond to leaf temperature. And leaf temperature in turn depends on plant water relations in addition to air temperature. Given the unique eco-hydrology of tundra plants, some attention should be paid to this driver. In particular, was soil moisture monitored for any of the chamber experiments? Was SM measured at the tower?

https://onlinelibrary.wiley.com/doi/pdf/10.1111/pce.12530

Minor comments Line 62: Delete second "to." Line 81: The term "flanks" is a bit odd. At least it should be singular. Line 86: Italicize "Vaccinium vitis-idaea." Line 109: very briefly give the details on the "moisture trap." Were cooled glass beads used? Line 110: What absorbents were used? Line 129: How large was this combined effect, in percent terms? Line 154: What uncertainty is introduced by data processing? Do you mean something related to statistics? Line 166: Please note when solar noon occurs at the site in AST. Line 179: Add a brief mention of how the tubes were capped. Lines 315-318: I concur with this conclusion. You could make this more clear and impactful by stating that both the intense wildfires regionally and the isoprene emissions locally were driven by high air temperature. But further, could there have been an influence on the photochemical lifetime of isoprene due to the products of the wildfire? Could the main isoprene oxidants, OH & ozone, be suppressed? Lines 319-335: Since you are integrating results and discussion, what's the implication of these results? Lines 338: There is no need to refer explicitly back to the Materials and Methods section, so "(see Section 2.3)" can be removed. Lines 343-345: What's the implication? Is the isoprene 'sticking around' from the more productive part of the day or is production continuing throughout the 'night' (low-PAR conditions). Lines 347-350: This is more than 'consistent.' I would change the wording to something along the lines of 'expected.'

Line 383: this is a _really_ high number and should be highlighted in the abstract. Unadjusted for temperature, biomass and light, it's similar to results from many mid- and low-latitude forests. Later in the paragraph, you give the comparison, which is good. But, I think the key is that the extreme values are so high. Line 406-409: Should explicitly state that even with nearly 24 hours of light, still get the typical diurnal pattern. The key is that low sun angles translate to very low PAR (non-linearly), and therefore you still see the typical diurnal pattern. Later in the paragraph, you get to this explicitly, but the discussion should be combined for clarity. Also, this should be related to the diurnal balloon experiment results. Lines 455-459: This needs to be tempered a bit. There are issues of timescales and acclimation. Also, I assume there are relatively few chamber measurements between 35 and 40 deg C, hence a leveling off is within statistical probabilities. Also, you will argue against this point in the following paragraph, so this could be presented more clearly to readers. Lines 461-462: I think I understand what this sentence is trying to convey, but it is confusing and the statement could be clearer. You that for every year in the dataset, there were 1-23 days with a temp above 20 deg C? Line 471: "Under" might be a better word chance than "scarcely." Line 472: Same comment as Line 81 about "flanks." Line 474: "Elevated" compared to what? Expectations or previous measurements? Line 477: Thermotolerance hasn't been addressed previously in the manuscript. At the minimum, a citation is necessary, but it might be best to remove this if it's not explored more thoroughly with regards to Arctic plants. Line 485: Can remove "likely" since "suggesting" is already in the sentence and provides sufficient caution. Line 486-490: Here thermotolerance is addressed a bit further, with references. But it would be better to have a short paragraph or group of sentences that speculates specifically about the role thermotolerance could play in promoting isoprene-emitting species in the Artic. The current allusions here and at Line 477 makes the topic appear as tacked on. Line 839, Table 1: some mention of the lack of measurements for R. chamaemorus would be useful, since it is the dominant species. Line 992, Figure 5: Solid, colored lines connecting the points would help visually highlight vertical trends.

---

## Author Comment (AC1) · 12 Oct 2020

Comments are in black and responses in blue.

**Response to Reviewer #1**

This paper aims at quantifying terpenoid mixing ratios and emission rates of dominant vegetation in northern Alaska. The authors have intensively compared their data with the published data from northern Sweden and Greenland and derived site-specific temperature response curve. This paper is well written and the data from this paper can provide base quantification of BVOC emissions from this less-studied area.

Thank you for the positive feedback. Our responses to the specific comments are provided below.

A variety of measurements have been used in this paper and I, as a modeler, will leave the measurement part to other reviewer(s). My main concern of this paper is that the mixing ratio measurements are not much linked to the emission rate measurements. It is a lot of data presented (which was good), but I think the authors should bring these data together to present a whole story.

We agree with the reviewer and have tried to better link the mixing ratio measurements to the emission rate measurements in the revised manuscript. We have, for instance, made the following additions:

"It is worth noting that the most frequently observed compounds in enclosure samples are among the most frequently seen MT in ambient air (see Section 3.1.3)".

"Regardless of the vegetation type, isoprene emission rates exhibited a significant diurnal cycle with an early afternoon maximum, in line with the mean diurnal cycle of enclosure temperature and PAR. These results are in line with the well-established diurnal variation of BVOC emissions in environments ranging from Mediterranean to boreal forests (e.g., Fares et al., 2013; Liu et al., 2004; Ruuskanen et al., 2005; Zini et al., 2001) and with the correlation between isoprene ambient air mixing ratios and temperature at TFS (see Section 3.1). (…) As can be seen in Table 3 and Fig. 8, PAR and BVOC emissions significantly decreased at night but were still detectable. These sustained BVOC emissions during nighttime confirm observations by Lindwall et al. (2015) during a 24-hour experiment with five different Arctic vegetation communities and explain the higher isoprene levels observed in the nocturnal boundary layer than above during the diurnal balloon experiment (see Section 3.1.2)."

Then, another part is about comparing emission ranges with literature values. The measurement conditions could vary largely and also in different periods of growing season. It is difficult to directly conclude that the measurement values are in the range of published values. I think standardized emission rates (using commonly-used Guenther algorithm) are needed in this case.

We totally agree and have used standardized emission rates (when possible/appropriate) in the revised manuscript (see below). We have also added the average enclosure temperature for each emission rate reported in Table 3.

"A branch enclosure experiment was performed from July 27 to August 2, 2018 on *Salix glauca* to investigate BVOC emission rates per dry weight plant biomass (see Fig.S.I.5). Isoprene emission rates ranged from <0.01 to 11 µgC/g/h (with a mean enclosure temperature of 16.5°C and mean PAR of 880 µmol/m$^2$/s), in line with non-normalized emission rates reported at Kobbefjord, Greenland by Kramshøj et al. (2016; Supplementary Table 5) for the same species under slightly different environmental conditions (mean temperature of 24.6°C and mean PAR of 1052 µmol/m$^2$/s). Once standardized to 30°C and 1000 µmol/m$^2$/s, our emission rates averaged 5 µgC/g/h, in good agreement with standardized emissions reported at Kobbefjord (mean of 7 µgC/g/h) by Vedel-Petersen et al. (2015)."

"The isoprene surface emission rate, as inferred from surface enclosures, was highly variable and ranged from 0.2 to ~2250 µgC/m$^2$/h (see Fig. 6). The 2250 µgC/m$^2$/h maximum, reached on June 26, 2019, with an enclosure temperature of 32°C, is higher than maximum values reported at TFS by Potosnak et al. (2013) (1200 µgC/m$^2$/h at an air temperature of 22°C). It should be noted that these maximum values were observed at different ambient temperatures; we further investigate the temperature dependency of isoprene emissions in Section 3.3. Elevated surface emission rates (*i.e.,* > 500 µgC/m$^2$/h) were all observed while the vegetation in sampling enclosures was dominated by *Salix* spp.. At TFS, the overall 24-hour mean isoprene emission rate amounted to 85 µgC/m$^2$/h, while the daytime (10 am-8 pm) and midday (11 am-2 pm) means were 140 and 213 µgC/m$^2$/h, respectively. To put this in perspective, the average isoprene surface emission rate standardized to 30°C and 1000 µmol/m$^2$/s (~ 300 µgC/m$^2$/h) was an order of magnitude lower than emission rates reported for warmer mid-latitude or tropical forests."

Here are some detailed comments:

Introduction: I would think one to two sentences could be needed to justify the importance of studying BVOC emissions on impacting atmospheric chemistry from

this less-polluted arctic region. Then I think the aim of this study should be elevated, so what are the main aim of this study apart from quantifying emissions and mixing ratios.

We thank the reviewer for these suggestions. We have made the following changes in the revised manuscript:

"Changing BVOC emissions in the Arctic due to climate and land cover shifts can thus be expected to perturb the overall oxidative chemistry of the region. Previous studies have hypothesized that BVOC might already impact the diurnal cycle of ozone in the Arctic boundary layer (Van Dam et al., 2016). Changing BVOC emissions can also further affect climate through various feedback mechanisms; Quantifying these changes requires an accurate understanding of the underlying processes driving BVOC emissions in the Arctic" (…). "The data presented here provide a baseline to investigate future changes in the BVOC emission potential of the under-studied Arctic tundra environment. Due to increasing shrub prevalence across northern Alaska, as well as the Eurasian and Russian Arctic, the results of this study have significance to tundra ecosystems across a vast region of the Arctic".

L52-53, the field warming increases of BVOC emission is not only seen with long-term warming but also found with a short-term field warming like 3 years in the same area.

We have replaced "Long-term field warming studies" by "Field warming studies" in the revised manuscript. Thank you for pointing that out.

L97, please describe the start and end of a normal growing season for this site.

We have clarified this in the revised manuscript: "These two back-to-back campaigns cover the entire growing season, from the onset of snow melt mid-May to the first snow fall mid-August".

Table1 Rhododendron tomentosum seems the 2nd highest covered in this area and why not present emission from this species separately?

*Rhododendron tomentosum* was indeed present in most of the surface enclosures but was difficult to study separately due to low individual plant biomass.

Fig. 3, I have a bit difficulty to find all measurements points at different heights. Suggest to use more distinguishing colors combining with different symbols.

We assume the reviewer actually refers to Figure 5 (vertical profiles with the tethered balloon). We have updated this Figure in the revised manuscript (see below; one panel per balloon flight) to make it easier to distinguish measurement points at different heights.

[Figure]

Figure 5: Vertical profiles of isoprene mixing ratios as inferred from 30-min samples collected with a tethered balloon. The error bars show the analytical uncertainty for isoprene (20 %). Samples with an isoprene mixing ratio lower than blanks were discarded. Hours are in Alaska Standard Time (UTC-9).

L362, as far as I know, Cassiope tetragona is also a MT emitter.

Thank you for this suggestion; *Cassiope tetragona* is included in "other high Arctic vegetation".

L372-L380, how valid it is to state that the values are in the range of published values if focusing on the emission rates potentially measured at very different temperature and light conditions. I would suggest comparing the standardized emission rates with other studies if possible.

See response to main comment #2.

L435, What does this mean "account for differing leaf area. . ."? This is the emission rate of the per ground area, right? Please clarify.

We have replaced "leaf area" by "total biomass" in the revised manuscript.

L436, If dividing all fluxes with the standard emission rates at 20 degree, then it gives a multiplication of environmental responses (unit-less). As PAR is measured in the chamber, why not take away the light variation part before only looking at temperature response curve? Then about Fig. 9, how did you deal with the MEGAN temperature response curve, as I did not see the normalized emission rate to 1 around 30 degree?

We followed the same methodology as in Tang et al. (2016) and only used daytime observations with relatively high PAR values. Figure 9 thus only represents the isoprene emission-temperature relationship. This has been clarified in the revised manuscript: "Figure 9 combines daytime (e.g., with relatively high PAR values) isoprene emission rates from different surface enclosures".
Please note that the MEGAN temperature response curve was also normalized by dividing all fluxes by the mean emission rate at 20°C.

L464-L468, MEGAN uses leaf temperature, not ambient air temperature for emission estimations. With predicted strong increase of air temperature in the Arctic, it still remains largely unknown (interesting to know) how plant leaf temperature can change and thus impact on BVOC emissions. I think it is important to have this in the discussion context.

Thank you for raising this important point. We agree that it is important to have this in the discussion and have added the following paragraph in Section 4.2 of the revised manuscript:

"Over the course of the two field campaigns at TFS, BVOC surface emission rates were measured over a large span of enclosure temperatures (2-41°C). While isoprene and MT emissions respond to leaf temperature (Guenther et al., 1993), air temperature was used here in place of leaf temperature – which has been assumed before in the literature for high-latitude ecosystems (e.g., Olofsson et al., 2005; Potosnak et al., 2013). Several studies have, however, suggested a decoupling of leaf and air temperature in tundra environments (Lindwall et al., 2016; Potosnak et al., 2013). With predicted increase of air temperature in the Arctic, it still remains largely unknown how leaf temperature will change and impact BVOC emissions. As suggested by Tang et al. (2016), long-term parallel observations of both leaf and air temperature are needed. The response of BVOC emissions to temperature discussed here should be interpreted with this potential caveat in mind."

---

## Author Comment (AC2) · 12 Oct 2020

Comments are in black and responses in blue.

**Response to Reviewer #2**

The manuscript presents results from a series of well-designed experiments that explore emissions of isoprene and monoterpenes from Artic plants to the atmosphere. The experimental design is sound and well-described, except for a few minor points noted below. The main conclusion is that the rapidly warming Arctic will cause sharply increased emissions of isoprene, which previous studies have shown to have a significant impact on atmospheric chemistry. Overall, this paper supports previous research findings, but the detail and atmospheric chemistry perspective make it a valuable contribution to the literature.

Thank you for the positive feedback. Our responses to the specific comments are provided below.

While the focus of the paper is from the atmosphere-exchange perspective, several eco-physiological concepts which have been discussed previously for Arctic BVOC emissions should be addressed. First, given the emphasis placed in the paper on the response of emissions to warming, the acclimation process should be addressed (see one reference below).

That is a very good point; thank you for raising it. We have added a new subsection in the revised manuscript (Section 4.2) to discuss long-term effects of warming, including the acclimation process:

"BVOC produced by plants are involved in plant growth, reproduction, and defense, and plants use isoprene emissions as a thermotolerance mechanism (Peñuelas and Staudt, 2010; Sasaki et al., 2007). The exponential response of isoprene emissions to temperature observed at TFS adds to a growing body of evidence indicating a high isoprene-temperature response in Arctic ecosystems. However, observations at TFS do not necessarily reflect long-term effects of warming. Schollert et al., (2015) examined how long-term warming affects leaf anatomy of individual arctic plant shoots (*Betula nana*, *Cassiope tetragona*, *Empetrum hermaphroditum*, and *Salix arctica*). They found that long-term warming results in significantly thicker leaves suggesting anatomical acclimation. While the authors hypothesized that this anatomical acclimation may limit the increase of BVOC emissions at plant shoot-level, Kramshøj et al. (2016) later showed that BVOC emissions from Arctic tundra exposed to six years of experimental warming increase at both the plant shoot and ecosystem levels.

In addition to the direct impact of long-term warming on BVOC emissions, ecosystem-level emissions are expected to increase in the Arctic due to climate-driven changes in plant biomass and vegetation composition. For instance, the widespread increase in shrub abundance in the Arctic – due to a longer growing season and enhanced nutrient availability (Berner et al., 2018; Sturm et al., 2001) – will likely significantly affect the BVOC emission potential of the Arctic tundra. Additionally, as mentioned above and as discussed extensively by Peñuelas and Staudt (2010) and Loreto and Schnitlzer (2010), emissions of BVOCs might be largely beneficial for plants, conferring them higher protection from abiotic stressors which are predicted to be more severe in the future. Long-term arctic warming may thus favor BVOC-emitting species even further."

Second, temperature in the current study refers to air temperature, but isoprene and other MT emissions respond to leaf temperature. And leaf temperature in turn depends on plant water relations in addition to air temperature. Given the unique eco-hydrology of tundra plants, some attention should be paid to this driver. In particular, was soil moisture monitored for any of the chamber experiments? Was SM measured at the tower?

We unfortunately did not monitor soil moisture but have made it clearer that the current study refers to air temperature:
"Over the course of the two field campaigns at TFS, BVOC surface emission rates were measured over a large span of enclosure temperatures (2-41°C). While isoprene and MT emissions respond to leaf temperature (Guenther et al., 1993), air temperature was used here in place of leaf temperature – which has been assumed before in the literature for high-latitude ecosystems (e.g., Olofsson et al., 2005; Potosnak et al., 2013). Several studies have, however, suggested a decoupling of leaf and air temperature in tundra environments (Lindwall et al., 2016; Potosnak et al., 2013). With predicted increase of air temperature in the Arctic, it still remains largely unknown how leaf temperature will change and impact BVOC emissions. As suggested by Tang et al. (2016), long-term parallel observations of both leaf and air temperature are needed. The response of BVOC emissions to temperature discussed here should be interpreted with this potential caveat in mind."

Line 62: Delete second "to."

Done, thanks for noticing this typo.

Line 81: The term "flanks" is a bit odd. At least it should be singular.

Done.

Line 86: Italicize "Vaccinium vitis-idaea."

Done.

Line 109: very briefly give the details on the "moisture trap." Were cooled glass beads used?

We have added a short description of the moisture trap in the revised manuscript: "The moisture trap was a U-shaped SilcoSteel$^{TM}$ tube (stainless steel treated) cooled using thermoelectric coolers". Note that the tube was empty (no glass beads).

Line 110: What absorbents were used?

This has been added to the revised manuscript: "Analytes were concentrated on a Peltier-cooled multistage micro-adsorbent trap (50 % Tenax-GR and 50 % Carboxen 1016)".

Line 129: How large was this combined effect, in percent terms?

We observed a progressive 80 % decline in CFC-113 peak area.

Line 154: What uncertainty is introduced by data processing? Do you mean something related to statistics?

The error introduced by data processing relates to the error in averaging the data from 2 minutes to 10 minutes, as well as error induced from peak fitting in the data processing software.

Line 166: Please note when solar noon occurs at the site in AST.

This has been added to the revised manuscript: "A total of eight vertical profiles were performed at ~3-hour intervals between 12:30 pm AST on June 15, 2019 and 11:00 am AST on June 16, 2019 in order to capture a full diurnal cycle (solar noon around 2 pm AST)".

Line 179: Add a brief mention of how the tubes were capped.

This has been added to the revised manuscript: "Following collection, adsorbent cartridges were sealed with Teflon-coated brass caps and stored in the dark at ~4°C until chemical analysis".

Lines 315-318: I concur with this conclusion. You could make this more clear and impactful by stating that both the intense wildfires regionally and the isoprene emissions locally were driven by high air temperature. But further, could there have been an influence on the photochemical lifetime of isoprene due to the products of the wildfire? Could the main isoprene oxidants, OH & ozone, be suppressed?

The occurrence of wildfires depends on meteorology (e.g., temperature and soil moisture) but also vegetation type and coverage, and lightning frequency. Fire emissions are a complicated mixture of trace gases and aerosols, many of which are short-lived and chemically reactive. This mixture affects the atmospheric composition in complex ways that are not completely understood. Recent measurements during the NASA/NOAA ATOM and FIREX-AQ campaigns have shown that wildfires might actually be responsible for increased ozone mixing ratios in aged plumes (Bourgeois et al., in prep). Our surface ozone measurements at Toolik Field Station suggest that mean ozone mixing ratios increased from ~27 ppb during June 1-19, 2019 to ~34 ppb during the June 20, 2019 fire event.

Lines 319-335: Since you are integrating results and discussion, what's the implication of these results?

We have added the following sentence in the revised version of the manuscript: "This record of ambient air isoprene, MACR, and MVK mixing ratios is, to the best of our knowledge, the first in an Arctic tundra environment. The combined measurement of isoprene and its oxidation products provides a new set of observations to further constrain isoprene chemistry under low-$NO_x$ conditions in atmospheric models (Bates and Jacob, 2019)."

Lines 338: There is no need to refer explicitly back to the Materials and Methods section, so "(see Section 2.3)" can be removed.

Done.

Lines 343-345: What's the implication? Is the isoprene 'sticking around' from the more productive part of the day or is production continuing throughout the 'night' (low-PAR conditions).

We now refer to Section 3.2.2: "Samples collected on June 16, 2019 from 4 to 4:30 am (see Fig. 5f) show decreasing isoprene mixing ratios with increasing elevation, suggesting higher levels (25-50 pptv) in the nocturnal boundary layer than above. This result suggests continuing isoprene emissions by the surrounding vegetation under low-PAR conditions. This is further discussed in Section 3.2.2".

Lines 347-350: This is more than 'consistent.' I would change the wording to something along the lines of 'expected.'

This has been modified in the revised manuscript: "This maximum at ground-level is expected for a VOC with a surface source (Helmig et al., 1998) while the 200 pptv mixing ratio can likely be attributed to a temperature-driven increase of isoprene emissions by the surrounding vegetation."

Line 383: this is a _really_ high number and should be highlighted in the abstract. Unadjusted for temperature, biomass and light, it's similar to results from many midland low-latitude forests. Later in the paragraph, you give the comparison, which is good. But, I think the key is that the extreme values are so high.

We agree and have highlighted this result in the abstract of the revised manuscript.

Line 406-409: Should explicitly state that even with nearly 24 hours of light, still get the typical diurnal pattern. The key is that low sun angles translate to very low PAR (non-linearly), and therefore you still see the typical diurnal pattern. Later in the paragraph, you get to this explicitly, but the discussion should be combined for clarity. Also, this should be related to the diurnal balloon experiment results.

We have clarified this in the revised manuscript: "Figures 8a-c show the mean diurnal cycle (over the two campaigns) of isoprene surface emission rates for different vegetation types. The two field campaigns were carried out during the midnight sun period, which could possibly sustain BVOC emissions during nighttime. It should, however, be noted that low sun angles translate to very low PAR and a typical diurnal pattern is observed in summer at TFS despite 24 hours of light (see Fig.8h)."

We have also related these results to the balloon vertical profiles: "These sustained BVOC emissions during nighttime confirm observations by Lindwall et al. (2015) during a 24-hour experiment with five different Arctic vegetation communities and explain the higher isoprene levels observed in the nocturnal boundary layer than above during the diurnal balloon experiment (see Section 3.1.2)."

Lines 455-459: This needs to be tempered a bit. There are issues of timescales and acclimation. Also, I assume there are relatively few chamber measurements between 35 and 40 deg C, hence a leveling off is within statistical probabilities. Also, you will argue against this point in the following paragraph, so this could be presented more clearly to readers.

We have added the number of chamber measurements in each temperature bin in the revised Figure 9. We do agree that, given the relatively few chamber measurements at T > 30°C, a leveling-off is within statistical probabilities. We have therefore tempered this paragraph accordingly in the revised manuscript (and in the abstract):

"While the model predicts a leveling-off of emissions at approximately 30-35°C, our observations reveal no such phenomenon within the 0-40°C enclosure temperature range (Fig. 9). However, given the limited number of enclosure measurements above 30°C, a leveling-off of emissions cannot be statistically ruled out. The key result here is that MEGAN2.1 adequately reproduces the temperature dependence response of Arctic ecosystems in the 0-30°C temperature range – ambient temperature > 30°C being unlikely."

Lines 461-462: I think I understand what this sentence is trying to convey, but it is confusing and the statement could be clearer. You that for every year in the dataset, there were 1-23 days with a temp above 20 deg C?

We have clarified this sentence in the revised manuscript: "Additionally, for each year in the 1988-2019 historical dataset, there were only 1 to 23 days (0 to 4 days) per year with a maximum temperature above 20°C (above 25°C)".

Line 471: "Under" might be a better word chance than "scarcely."

The wording has been changed accordingly in the revised manuscript.

Line 472: Same comment as Line 81 about "flanks."

Done.

Line 474: "Elevated" compared to what? Expectations or previous measurements?

This has been clarified in the revised manuscript: "While the overall mean isoprene emission rate amounted to 85 µgC/m$^2$/h at TFS, elevated (> 500 µgC/m$^2$/h) isoprene surface emission rates were observed for *Salix* spp., a known isoprene emitter."

Line 477: Thermotolerance hasn't been addressed previously in the manuscript. At the minimum, a citation is necessary, but it might be best to remove this if it's not explored more thoroughly with regards to Arctic plants.

Thermotolerance is now addressed in Section 4.2 of the revised manuscript (see above).

Line 485: Can remove "likely" since "suggesting" is already in the sentence and provides sufficient caution.

Done.

Line 486-490: Here thermotolerance is addressed a bit further, with references. But it would be better to have a short paragraph or group of sentences that speculates specifically about the role thermotolerance could play in promoting isoprene-emitting species in the Artic. The current allusions here and at Line 477 makes the topic appear as tacked on.

This is now addressed in Section 4.2 of the revised manuscript (see above).

Line 839, Table 1: some mention of the lack of measurements for R. chamaemorus would be useful, since it is the dominant species.

Indeed. This has been added in the revised manuscript: "The tundra vegetation around TFS is heterogeneous but most dominant species (except *Rubus chamaemorus*) were sampled."

Line 992, Figure 5: Solid, colored lines connecting the points would help visually highlight vertical trends.

We have updated this Figure in the revised manuscript (see below; one panel per balloon flight) to make it easier to distinguish measurement points at different heights.

[Figure]

Figure 5: Vertical profiles of isoprene mixing ratios as inferred from 30-min samples collected with a tethered balloon. The error bars show the analytical uncertainty for isoprene (20 %). Samples with an isoprene mixing ratio lower than blanks were discarded. Hours are in Alaska Standard Time (UTC-9).

---

## Author Response (AR2)

Comments are in black and responses in blue.

**Response to Reviewer #3**

Angot and others measure biogenic volatile organic compound fluxes and atmospheric concentrations in a tundra environment. The measurements were carried out competently and the study is interesting, my comments are only minor.

Line 36: some species have even shorter lifetimes e.g. https://agupubs.onlinelibrary.wiley.com/doi/abs/10.1029/95JD00368

We have revised this sentence accordingly: "Despite their relatively short atmospheric lifetimes (a few minutes to 1 day for terpenoids) (...)".

Line 177: Was the 30-minute sampling period in order to ensure that enough material was collected? There will be vertical mixing between levels at that time scale, even in the arctic, especially because sensible heat flux and convective energy is higher than many people realize due to the inefficiency of mosses and lichens at moving water vapor to the atmosphere (e.g. https://bg.copernicus.org/articles/8/3375/2011/).

Indeed, the manuscript has been revised accordingly: "Once the balloon reached its apex (~250-300 m a.g.l.), the five pumps were activated simultaneously and samples collected for 30 minutes to ensure that enough material was collected. It should be noted that changes in wind speed and turbulence during the 30-min sampling period often affected the shape of the tethered line and the sampling altitude adding further uncertainty to the vertical profiles presented here".

('Clock' also probably shouldn't be capitalized).

Done.

I'm not really sure what 'Miscellaneous' means in Figure 8. Perhaps it was mentioned in the text but the legend did not define it. I found the used on 216. More explanation of what is included in this grouping (which might accurately be called 'other') because it is not a 'vegetation type' (line 215) would lead to less confusion. Does it combine vascular and non-vascular species for example?

The definition of 'Miscellaneous' (mix of different species, including lichens and moss tundra) has been added to the caption of Figures 6, 7, and 8.

**1 Biogenic volatile organic compound ambient mixing ratios and emission rates**

**2 in the Alaskan Arctic tundra**

- 3 Hélène Angot1, Katelyn McErlean1, Lu Hu2, Dylan B. Millet3, Jacques Hueber1, Kaixin Cui1, Jacob Moss1,
- 4 Catherine Wielgasz2, Tyler Milligan1, Damien Ketcherside2, Marion Syndonia Bret-Harte4, Detlev Helmig1

[revised manuscript text omitted]

|                          | Relative land surface |                       |  |
|--------------------------|-----------------------|-----------------------|--|
| Dlaut name               | cover in moist acidic | Present in surface or |  |
| Plant name               | tundra (%) (Gough,    | bag enclosures        |  |
|                          | 2019)                 |                       |  |
| Andromeda polifolia      | 0.6                   | yes                   |  |
| Betula nana              | 14.4                  | yes                   |  |
| Carex bigelowii          | 1.0                   | yes                   |  |
| Cassiope tetragona       | 2.0                   | yes                   |  |
| Empetrum nigrum          | 3.8                   | yes                   |  |
| Eriophorum
vaginatum  | 8.6                   | yes                   |  |
| Ledum palustre           | 10.5                  | yes                   |  |
| Mixed Lichens            | 2.1                   | yes                   |  |
| Mixed moss               | 6.0                   | yes                   |  |
| Pedicularis
lapponica | 0.6                   | no                    |  |
| Polygonum bistorta       | 0.6                   | no                    |  |
| Rubus chamaemorus        | 20.2                  | no                    |  |
| Salix pulchra            | 4.9                   | yes                   |  |
| Vaccinium
uliginosum  | 1.9                   | yes                   |  |
| Vaccinium vitis-idaea    | 6.6                   | yes                   |  |

- 924 Table 2: Average mixings ratios with standard deviation, along with minimum (min) and maximum (max)
- 925 values and quantification frequency (QF) of the measured monoterpenes in ambient air. LOQ stands for
- 926 limit of quantification. For values lower than the LOQ, mixing ratios equal to half of the LOQ were used
- 927 to calculate the mean.

|          | mean ± standard deviation
(pptv) | Min (pptv) | Max (pptv) | QF (%) |
|----------|-------------------------------------|------------|------------|--------|
| α-pinene | $11.7 \pm 8.1$                      | < LOQ      | 61.6       | 88     |
| camphene | < LOQ                               | < LOQ      | 21.9       | 11     |
| sabinene | < LOQ                               | < LOQ      | 34.2       | 11     |
| p-cymene | 2.0 ± 1.9                           | < LOQ      | 12.3       | 32     |
| limonene | < LOQ                               | < LOQ      | 2.9        | < 1    |

- -

942 Table 3: Isoprene and monoterpenes (sum of  $\alpha$ -pinene,  $\beta$ -pinene, limonene, and 1,8-cineole) surface

943 emission rates per vegetation type. Miscellaneous refers to a mix of different species, including lichens and

944 moss tundra (see Fig.S.I.3-15). Daytime refers to 10 am-8 pm, midday to 11 am-2 pm, and nighttime to 11

945 pm-5 am (Alaska Standard Time). The values in brackets represent the average enclosure temperature for

946 each emission rate.

|               | $mean \pm standard$ | daytime mean $\pm$ | midday mean $\pm$  | nighttime mean ±   |  |  |
|---------------|---------------------|--------------------|--------------------|--------------------|--|--|
|               | deviation           | standard deviation | standard deviation | standard deviation |  |  |
|               | $(\mu gC/m^2/h)$    | $(\mu gC/m^2/h)$   | $(\mu gC/m^2/h)$   | $(\mu gC/m^2/h)$   |  |  |
| isoprene      |                     |                    |                    |                    |  |  |
| Salix spp.    | $149\pm327$         | $232\pm400$        | $334\pm473$        | 7 ± 10             |  |  |
|               | [17.6°C]            | [23.9°C]           | [27.0°C]           | [8.0°C]            |  |  |
| Betula spp.   | $12 \pm 30$         | $19 \pm 38$        | $28 \pm 37$        | 5 ± 14             |  |  |
|               | [13.7°C]            | [17.4°C]           | [20.1°C]           | [5.8°C]            |  |  |
| Miscellaneous | $38 \pm 81$         | $57 \pm 100$       | $104 \pm 135$      | 21 ± 64            |  |  |
|               | [11.8°C]            | [14.8°C]           | [16.2°C]           | [8.2°C]            |  |  |
| monoterpenes  |                     |                    |                    |                    |  |  |
| Salix spp.    | 0.8 ± 1.3           | 1.1 ± 1.5          | $1.4 \pm 1.7$      | 0.4 ± 1.0          |  |  |
|               | [17.6°C]            | [23.9°C]           | [27.0°C]           | [8.0°C]            |  |  |
| Betula spp.   | $0.5 \pm 0.6$       | $0.7 \pm 0.7$      | $1.0 \pm 0.8$      | $0.2 \pm 0.2$      |  |  |
|               | [13.7°C]            | [17.4°C]           | [20.1°C]           | [5.8°C]            |  |  |
| Miscellaneous | $1.1 \pm 1.4$       | $1.3 \pm 1.6$      | $1.7 \pm 2.0$      | 1.0 ± 1.4          |  |  |
|               | [11.8°C]            | [14.8°C]           | [16.2°C]           | [8.2°C]            |  |  |

947

948

~ т(

949

950

951

954 Figure 1: Location of Toolik Field Station (TFS) on the north flanks of the Brooks Range in northern Alaska

along with arctic vegetation type. This Figure was made using the raster version of the Circumpolar Arctic

956 Vegetation Map prepared by Raynolds et al. (2019) and publicly available at www.geobotany.uaf.edu.